# Global cellular response to chemotherapy-induced apoptosis

Arun P Wiita[1,2], Etay Ziv[3,4], Paul J Wiita[5], Anatoly Urisman[1,6], Olivier Julien[1], Alma L Burlingame[1], Jonathan S Weissman[3,7], James A Wells[1,3]*

[1]Department of Pharmaceutical Chemistry, University of California, San Francisco, San Francisco, United States; [2]Department of Laboratory Medicine, University of California, San Francisco, San Francisco, United States; [3]Department of Cellular and Molecular Pharmacology, University of California, San Francisco, San Francisco, United States; [4]Department of Radiology, University of California, San Francisco, San Francisco, United States; [5]Department of Physics, The College of New Jersey, Ewing, United States; [6]Department of Pathology, University of California, San Francisco, San Francisco, United States; [7]Howard Hughes Medical Institute, University of California, San Francisco, San Francisco, United States

**Abstract** How cancer cells globally struggle with a chemotherapeutic insult before succumbing to apoptosis is largely unknown. Here we use an integrated systems-level examination of transcription, translation, and proteolysis to understand these events central to cancer treatment. As a model we study myeloma cells exposed to the proteasome inhibitor bortezomib, a first-line therapy. Despite robust transcriptional changes, unbiased quantitative proteomics detects production of only a few critical anti-apoptotic proteins against a background of general translation inhibition. Simultaneous ribosome profiling further reveals potential translational regulation of stress response genes. Once the apoptotic machinery is engaged, degradation by caspases is largely independent of upstream bortezomib effects. Moreover, previously uncharacterized non-caspase proteolytic events also participate in cellular deconstruction. Our systems-level data also support co-targeting the anti-apoptotic regulator HSF1 to promote cell death by bortezomib. This integrated approach offers unique, in-depth insight into apoptotic dynamics that may prove important to preclinical evaluation of any anti-cancer compound.

*For correspondence: jim.
wells@ucsf.edu

Competing interests: The authors declare that no competing interests exist.

## Introduction

Most cytotoxic chemotherapeutics eliminate tumor cells by activating the intrinsic apoptotic pathway (*Kaufmann and Earnshaw, 2000*). The final stages leading to mitochondrial pore formation and caspase activation have been well-described at the molecular level (*Gonzalvez and Ashkenazi, 2010*; *Spencer and Sorger, 2011*; *Parrish et al., 2013*). However, prior to these terminal stages of apoptosis, it is becoming clear that cells fight to resist the pharmacological insult in ways that appear unique to the treatment applied (*Geva-Zatorsky et al., 2010*). Existing genome-wide studies of cellular response to chemotherapeutic treatment have primarily relied on changes at the transcriptional level (*Lamb et al., 2006*). Elegant single cell studies have tracked a subset of proteins in response to various chemotherapeutics (*Cohen et al., 2008*; *Geva-Zatorsky et al., 2010*). Recent work has identified hundreds of caspase cleavage substrates in apoptotic cells treated with chemotherapeutics (*Shimbo et al., 2012*).

However, each of these approaches only captures a segment of the functional reaction to a chemotherapeutic insult and does not tell the full story of how cancer responds to apoptosis-inducing drug treatment. It is known that cells undergoing apoptosis show strong suppression of protein translation (*Bushell et al., 2004*). While a few specific transcripts are known to escape this translational suppression (*Spriggs et al., 2010*), the general link between transcriptional and translational changes during apoptosis

**eLife digest** Many cancer treatments work by causing cancer cells to enter an advanced stage of a process known as programmed cell death or apoptosis. When a cell begins apoptosis, it takes a series of metabolic steps–such as fragmenting its DNA or reducing its volume–that eventually kills it. The cancer cells in tumours are able to grow because they are able to avoid apoptosis.

When cancer cells are treated with cytotoxic drugs they do not die immediately but try to stave off the effect of the drug. However, we still know relatively little about what happens at the molecular levels as cancer cells struggle to avoid apoptosis.

Now Wiita et al. have combined two methods for studying cancer cells–deep sequencing of RNA and quantitative proteomics–to simultaneously observe a variety of processes, including the transcription of genes to produce messenger RNA (mRNA) molecules, the translation of these mRNA molecules to produce proteins, and the proteolysis (or breakdown) of these proteins when the cells were subjected to chemotherapy.

Wiita et al. studied how human myeloma cells responded to bortezomib, a drug that is used to treat various blood cancers, and found that ribosomes–the complex molecular machines that perform the translation step– reacted to the chemotherapy by preferentially translating certain mRNA molecules in order to produce a set of proteins that protect the cell. Developing drugs to inhibit the effects of these stress-response proteins could make the cancer cells more responsive to existing anticancer drugs. When this effort to stay alive is ultimately unsuccessful, the destruction of proteins appears surprisingly unrelated to the previous attempts that were made to protect the cell.

With further work the "global cellular response" approach developed by Wiita et al. could lead to the discovery of new drug targets, improve our understanding of drug resistance in chemotherapy, and provide new ways to monitor how patients respond to treatment.

is not well understood. Furthermore, different chemotherapeutics produce distinct quantitative signatures of caspase cleavage substrates following apoptosis (*Shimbo et al., 2012*), and it is unclear how the cellular response to chemotherapy prior to apoptotic induction may influence the later deconstruction of cellular protein contents.

Here we examine in depth the response of a multiple myeloma cell culture line (MM1.S) exposed to the proteasome inhibitor bortezomib, a clinically relevant model system. Multiple myeloma, one of the most common blood cancers, is an aggressive malignancy of clonal plasma cells. Proteasome inhibition has become an effective first-line therapy for this disease, though myeloma currently has no known cure (*Lonial et al., 2011*). We use a suite of emerging systems-level approaches to globally examine the dynamic interplay between transcription, translation, and proteolytic degradation during chemotherapy-induced apoptosis in this system (*Figure 1A*). Via ribosome profiling, we identify preferential translation and translational regulation of genes expected to reduce unfolded protein stress after bortezomib treatment. Surprisingly, despite these changes in translational control, during rapid apoptosis only a few critical pro-survival proteins are detectably increased in proteomic studies. Furthermore, we find that cleavage patterns by both caspase and non-caspase proteases during apoptosis are largely independent of drug response at the transcriptional and translational level prior to cell death. We also use this integrated data to examine the potential of small molecule therapeutics to work in concert with bortezomib in myeloma. Such a global examination of how cells struggle at all levels with a chemotherapeutic insult provides important insights into mechanisms of therapeutic resistance and novel methods to assess tumor response to chemotherapy.

## Results

### Simultaneous monitoring of multiple systems-level processes during apoptosis

We designed our experiment to monitor four systems-level processes simultaneously in the same cellular population, from transcription through protein production and proteolysis (*Figure 1A*). Deep sequencing of mRNA (mRNA-seq) examines the detailed transcriptional response of myeloma cells to

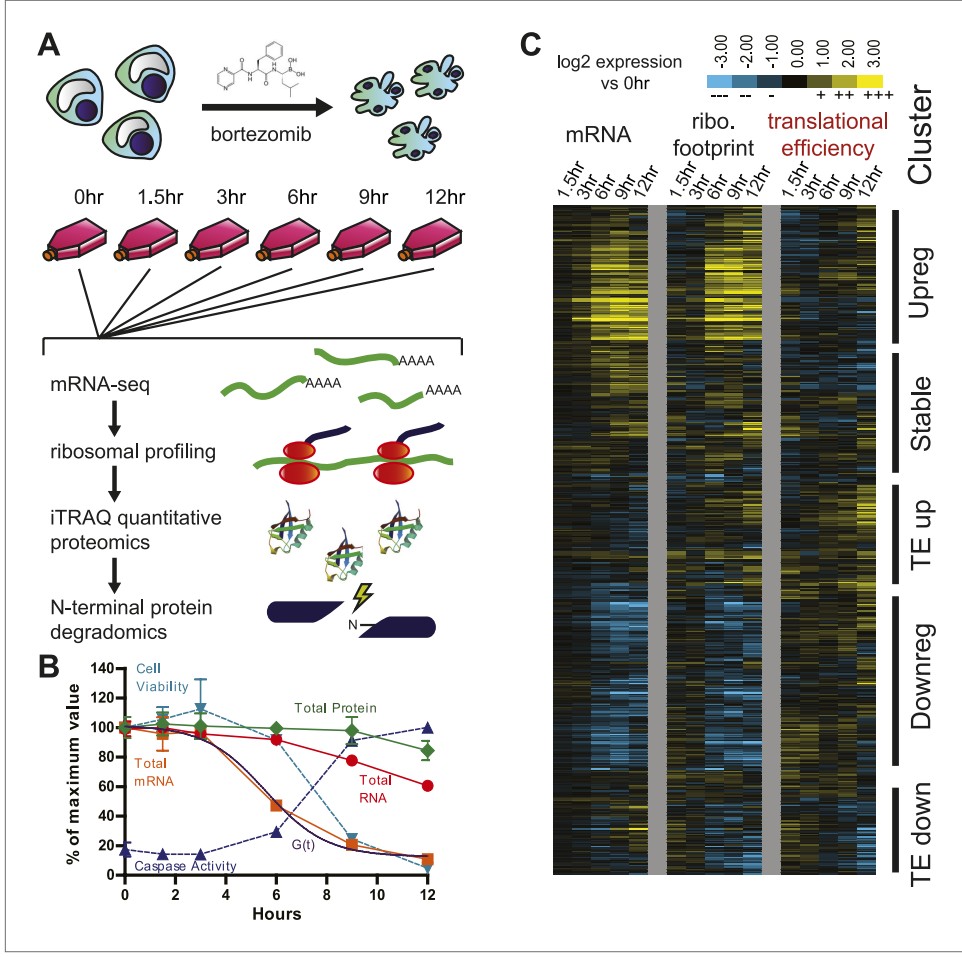

**Figure 1**. Experimental design and paired mRNA-seq/ribosome profiling data. (**A**) MM1.S cells were prepared in six separate flasks each containing $3 \times 10^8$ cells. All time points past 0 hr were exposed to 20 nM bortezomib. Cells were harvested and analyzed using four systems-level technologies shown. (**B**) Cells undergo rapid apoptosis with initial increase in caspase activation seen at 6 hr and <10% cell viability by 12 hr (duplicate, mean ± SD). Total RNA and mRNA were isolated and measured by spectrophotometry; total protein was measured by BCA assay (measured in duplicate, isolated from $10^6$ cells; mean ± SD). G($t$), describing global mRNA degradation, is derived from a sigmoid fit to the total mRNA data. (**C**) Heat map showing $\log_2$ expression compared to 0 hr for normalized mRNA and ribosome footprint read density (reads per kilobase million, RPKM) for 5680 well-expressed transcripts (*Figure 1— source data 1*). Also shown is calculated translational efficiency (TE) relative to 0 hr, calculated as the ratio of (footprint RPKM)/(mRNA RPKM) per transcript at each time point. We used unsupervised hierarchical clustering to define five broad groups of transcripts with relation to mRNA, footprint, and translational efficiency changes.

The following source data and figure supplements are available for figure 1:

**Source data 1**. Deep sequencing data, translational efficiency, and gene clusters.

**Figure supplement 1**. Sample comparison and biological subgroups.

**Figure supplement 2**. Immunoblots.

**Figure supplement 3**. Biological subgroups.

bortezomib-induced stress. We pair this data with ribosome profiling, the deep sequencing of ribosome-protected mRNA fragments during active translation; this approach offers significant insight into proteome dynamics and translational regulation not available by monitoring transcript expression alone (*Ingolia et al., 2009*, *2011*; *Stern-Ginossar et al., 2012*). To directly measure changes at the protein level after

bortezomib perturbation, we use isobaric labeling by iTRAQ (*Mertins et al., 2012*) and tandem mass spectrometry to measure in an unbiased fashion the relative abundance of ~2600 proteins. Finally, using enzymatically-driven labeling of free protein α-amines (N-terminomics) (*Mahrus et al., 2008*; *Agard et al., 2012*), we develop a quantitative mass spectrometry assay to monitor to the relative kinetics of both caspase and non-caspase cleavage events during apoptotic cellular deconstruction. To begin, we prepared separate flasks of MM1.S cells and harvested cells at five time points after induction of apoptosis with 20 nM bortezomib. Similar to previous studies (*Shimbo et al., 2012*), caspase activity was first detected at 6 hr and the majority of cells were non-viable by 12 hr (*Figure 1B*). mRNA levels were also markedly decreased prior to loss of cell viability, as previously seen across a range of apoptotic conditions (*Del Prete et al., 2002*; *Bushell et al., 2004*). Total RNA levels, primarily reflecting ribosomal RNA abundance, and total protein levels were relatively stable.

## Ribosome occupancy generally tracks with transcript production

To understand how myeloma cells were struggling after bortezomib exposure, we determined the relative changes in transcription and translation within the remaining mRNA pool at each time point. We normalized mRNA-seq and ribosome profiling sample using reads per kilobase million (RPKM) and focused our analysis on 5680 well-expressed transcripts (*Figure 1C*). We compared $\log_2$ expression vs the untreated sample and organized the data using unsupervised hierarchical clustering. This approach allowed us to center on relative expression changes found in multiple independent samples. We generally found that changes in ribosome footprints tracked with changes in transcript abundance. The transcripts could be sub-divided into five clusters (*Figure 1C*). Clusters 'Upreg' and 'Downreg' encompass transcripts that are generally increased or decreased, respectively, at both the level of mRNA and footprint reads (*Figure 2A,B*). Cluster 'Stable' includes transcripts that demonstrate very mild changes at both the mRNA and footprint level. We calculated the translational efficiency (TE) from the ratio of ribosome foot-print to mRNA-seq read density for each transcript (*Ingolia et al., 2011*). This enabled us to characterize two other groups that are particularly interesting, Cluster 'TE Up' and 'TE Down'. These groups showed little change in the level of mRNA, but large increases or decreases, respectively, in relative ribosome foot-print density. This suggests that genes in these two clusters are regulated at the level of translation.

## A subset of stress response genes show changes in translational efficiency

Bortezomib is known to be an inducer of the unfolded protein response (UPR) and endoplasmic reticulum (ER) stress (*Aronson and Davies, 2012*). Myeloma plasma cells are particularly sensitive to proteasome inhibition in vivo due to extremely high rates of immunoglobulin production, potentially leading to apoptosis via the UPR (*Obeng et al., 2006*; *Walter and Ron, 2011*). In *Figure 2C* we display reads for one of the critical factors in the UPR, *ATF4,* a transcription factor known to be under translational control during ER stress (*Lu et al., 2004*). Consistent with this prior work, we find that *ATF4* does not increase in transcript abundance but shows a nearly threefold increase of ribosome occupancy (see heat map in *Figure 1—figure supplement 3*).

To better assess the biological implications of these results we turned to Ingenuity Pathway Analysis (IPA) (Ingenuity Systems, www.ingenuity.com) (*Table 1*). Cluster Upreg is enriched for genes related to protein ubiquitination (p=$1.50 \times 10^{-34}$), protein degradation ($9.63 \times 10^{-9}$), chaperones ($4.15 \times 10^{-11}$), and hypoxic response ($2.90 \times 10^{-10}$). Cluster Downreg includes genes important in cellular proliferation (p=$3.61 \times 10^{-11}$) and DNA repair ($3.81 \times 10^{-9}$). These findings are consistent with mRNA microarray studies of bortezomib response (*Mitsiades et al., 2002*). We examined in more detail a subset of genes related to cellular apoptosis and both ER stress and hypoxic response (*Figure 1—figure supplement 3*). Notably, we found little change in expression or translation of canonical apoptosis players.

Surprisingly, IPA showed that Cluster TE Up included many genes regulated by XBP1 (p=$8.79 \times 10^{-10}$) (*Table 1*), an important component of the UPR (*Walter and Ron, 2011*). The only known function of XBP1 is as a transcription factor (*Acosta-Alvear et al., 2007*) without known direct effects on translation. Although we do see a slight increase in the active XBP1 protein (*Figure 1—figure supplement 2*), most downstream XBP1 targets do not respond by increases in mRNA abundance. Instead we see increased translation efficiency (*Figure 2—figure supplement 1*). Notably, transcripts related to the ER transport machinery (including *SEC61B, Figure 2D*) demonstrate 1.5- to 2-fold increases in translation vs transcription. Therefore, changes in translation of XBP1downstream targets may reflect the activity of a parallel UPR mechanism to favor adaptation to cellular stress.

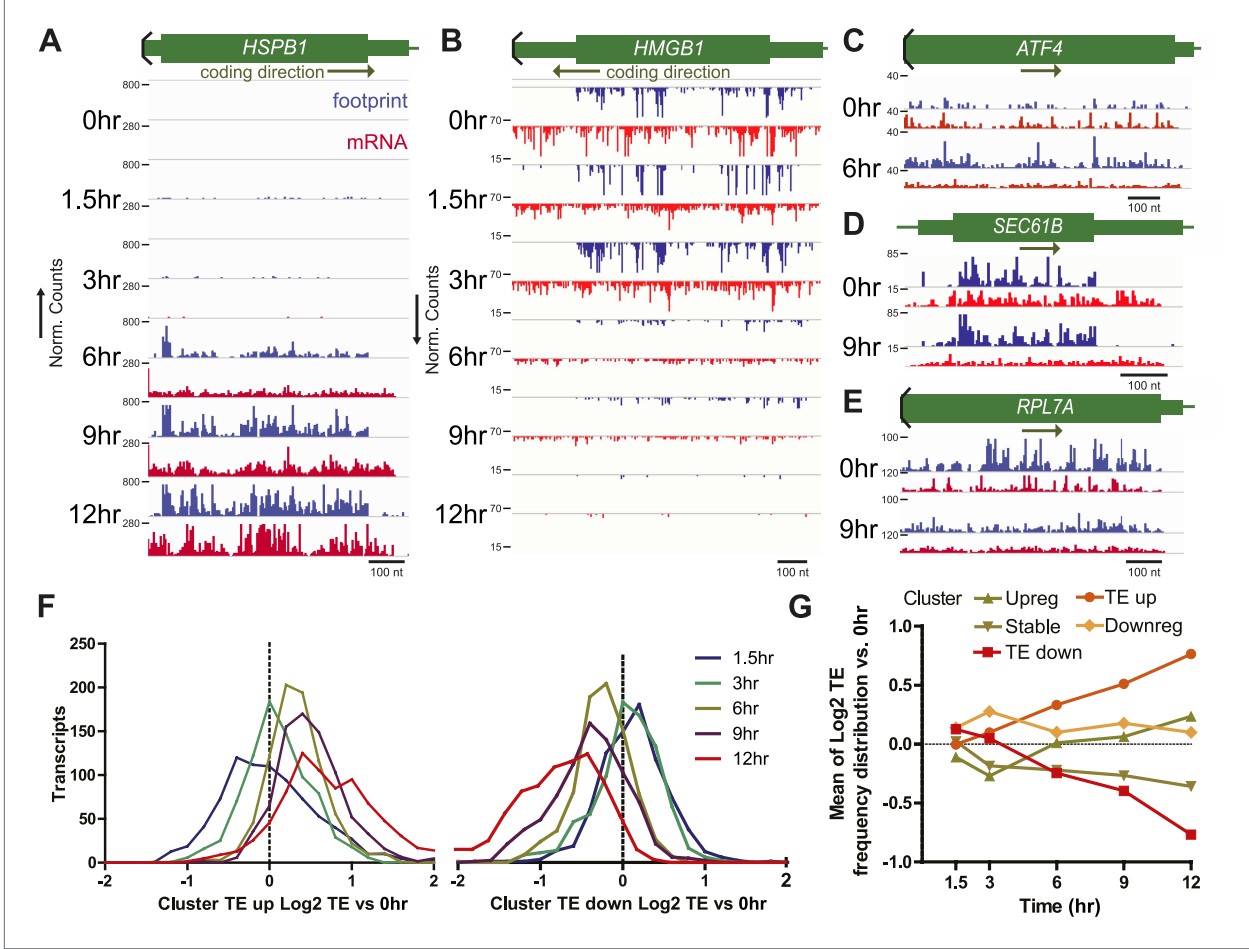

**Figure 2**. Translational efficiency (TE) during bortezomib-induced apoptosis. (**A** and **B**) Paired mRNA and ribosome footprint reads for *HSPB1* (Cluster Upreg) and *HMGB1* (Cluster Downreg) showing that protein translation generally tracks with transcript abundance. Displayed read counts are median-normalized to total aligned reads in 0 hr sample for comparison across the time course. Thick green bars = protein coding sequence (CDS); medium bars = untranslated regions (UTR); thin lines = intronic or intergenic regions. Read counts are inverted depending on coding direction. (**C**) *ATF4* shows increased footprint read density in the absence of mRNA increase at 6 hr. (**D**) *SEC61B*, a downstream target of XBP1, shows increased footprints and decreased mRNA reads, leading to increased TE. (**E**) Many ribosome structural proteins, including *RPL7A*, are found in Cluster TE down. (**F**) $Log_2$ changes in transcript TE from Cluster TE up and Cluster TE down at 12 hr vs 0 hr demonstrate statistically significant changes in the overall TE distribution (p<0.0001, Mann-Whitney test). (**G**) Mean (±SEM) changes in TE during the apoptotic time course generally do not exceed twofold even at later time points.

The following figure supplements are available for figure 2:

**Figure supplement 1**. Subgroup analysis from biological pathways showing changes in translation efficiency from Ingenuity Pathway Analysis.

Cluster TE Down includes many transcripts involved in the translational machinery itself, particularly ribosomal structural proteins (*Figure 2E*, *Figure 2—figure supplement 1*) of which the most significant pathway affected involved eIF2 signaling (p=$2.35 \times 10^{-17}$) (*Table 1*). In parallel we examined the 5% of genes with the highest translational efficiency in the untreated sample. The pathway 'mitochondrial dysfunction' was highly enriched (p=$1.52 \times 10^{-26}$) (*Table 1*), including numerous cytochrome c oxidase and NADH dehydrogenase subunits (*Figure 1—source data 1*). This finding suggests a favoring of translation of aerobic metabolism components in myeloma cells at baseline. In *Figure 2F* we display the distribution of $log_2$-fold changes in translational efficiency for Clusters TE Up and TE Down, which both show a significant difference between 12 hr vs 1.5 hr (p<0.0001, Mann-Whitney test). These changes contrast with the large majority of transcripts which show little change in TE (*Figure 2G*).

**Table 1.** Biological relevance of findings from Ingenuity Pathway Analysis (IPA)

| Cluster | Molecular and cellular function | p value | Canonical pathway | p value | Upstream regulator | p value |
|---|---|---|---|---|---|---|
| Cluster Upreg | Post-translational modification | 4.15 e-11 | Protein ubiquitination pathway | 1.50e-34 | NFE2L2 (NRF2) | 7.77e-16 |
| | Protein Folding | 4.15 e-11 | NRF2-mediated oxidative stress response | 2.90e-10 | HSF1 | 1.70e-14 |
| | Cell death and survival | 2.47 e-10 | | | EIF2AK3 (PERK) | 6.47e-10 |
| | Protein synthesis | 8.82 e-09 | | | | |
| | Protein degradation | 9.63 e-09 | | | | |
| Cluster Stable | RNA post-transcriptional modification | 1.44e-09 | – | | – | |
| | Molecular transport | 8.87 e-09 | | | | |
| | Protein trafficking | 8.87 e-09 | | | | |
| Cluster TE Up | – | | Mitochondrial dysfunction | 6.04e-10 | XBP1 | 8.79e-10 |
| Cluster Downreg | Cell death and survival | 7.56e-12 | – | | TP53 | 2.67e-15 |
| | Cellular growth and proliferation | 3.61e-11 | | | MYC | 1.53e-10 |
| | DNA replication, recombination and repair | 3.81e-09 | | | XBP1 | 1.65e-10 |
| | RNA post-transcriptional modification | 8.91e-08 | | | INSR | 5.94e-09 |
| | | | | | E2F4 | 6.20e-09 |
| Cluster TE Down | RNA post-transcriptional modification | 8.88e-11 | EIF2 signaling | 2.35e-17 | – | |
| | Gene expression | 3.24e-09 | Regulation of eIF4 and p70S6K signaling | 1.40e-08 | | |
| 5% transcripts highest TE at 0 hr | – | | Mitchondrial dysfunction | 1.52e-26 | – | |
| Increased 5′ UTR translation | – | | EIF2 signaling | 2.45 e-17 | MYC | 1.09e-08 |
| | | | | | HSF1 | 8.11e-08 |
| Decreased 5′ UTR translation | – | | – | | – | |

Clusters defined by hierarchical clustering of mRNA, footprint, and translational efficiency data as shown in **Figure 1B** (included genes listed in **Figure 1—source data 1**). We also analyzed the top 5% of transcripts by translational efficiency at 0 hr as calculated by (footprint RPKM)/(mRNA RPKM) per transcript. p values are as calculated using Fisher's exact test by Ingenuity Pathway Analysis software. We report findings here with a $p < 10^{-7}$. Results from each category (Molecular and cellular function; Canonical pathway; Upstream regulator) are listed in order of decreasing p, independent of the other categories.

## Differential 5′ UTR translation

Taking advantage of the nucleotide resolution of ribosome profiling, we examined whether there were any large-scale changes in ribosome occupancy along mRNA during apoptosis. We performed a meta-gene analysis, where all footprint profiles are averaged and then aligned based on the midpoint of the

protected reads (*Figure 3A*). We find a strong peak of ribosome occupancy at both the 5′ and 3′ ends of annotated coding sequence (CDS) and a 3-nucleotide offset resulting from position of the ribosome P site, as described previously (*Ingolia et al., 2009*). We also note peaks appearing every three nucleotides across the averaged reads, consistent with the triplet nucleotide coding sequence. Averaged across all transcripts, we did not find any large changes in footprint read distribution across mRNAs at different time points.

We next investigated whether there were general changes in proteome control by differing ribosome occupancy of the mRNA 5′ untranslated region (UTR). Ribosome occupancy in this region may indicate translation of short regulatory polypeptides in upstream open reading frames (uORFs) or production of alternate N-terminal isoforms of canonically translated proteins (*Ingolia et al., 2011*; *Lee et al., 2012*; *Stern-Ginossar et al., 2012*). Others have shown that yeast responding to oxidative stress produce large increases in 5′ UTR translation (*Gerashchenko et al., 2012*). A general decrease in 5′ UTR translation was also seen in differentiating mouse embryonic stem cells (*Ingolia et al., 2011*). In contrast, we identified no overall trend toward altered 5′ UTR translation relative to CDS translation in our system (*Figure 3B*). However, these frequency distributions are broad and some individual transcripts do have large changes in the relative translation of the 5′ UTR during apoptosis. We examined transcripts with >twofold change in 5′ UTR translation relative to CDS translation when compared to untreated sample in at least three time points after drug exposure (*Figure 3—source data 1*). By $\chi^2$ analysis, genes in Cluster Upreg were significantly over-represented among the 274 genes in the increased UTR translation group (p=0.033). Both Clusters Upreg and Downreg were over-represented among the 219 genes in the decreased UTR translation group (p=0.0006 and p=0.014, respectively). No other Clusters showed significant over- or under-representation. We did not find any difference in 5′ UTR length between the groups with most increased and most decreased relative 5′ UTR translation (*Figure 3—figure supplement 1*). IPA (*Table 1*) and inspection of genes in the increased UTR translation group revealed numerous translation elongation and initiation factors, ribosome structural proteins, chaperones (i.e., *DNAJA1*, *HSP90AB1*, *CCT5*, *HSPA8*; *Figure 3C*), and proteasomal subunits. The decreased UTR translation group does not show as strong a biological pattern although we do find ubiquitin (*UBB*), the co-chaperone BAG3, and cytochrome *c* (*CYCS*; *Figure 3D*), that are relevant to bortezomib-induced apoptosis.

The production of ATF4 is thought to be governed by the relative translation of two uORFs in the 5′ UTR (*Lu et al., 2004*). We find that changes in the footprint density across both of these uORFs correspond with increased translation of the *ATF4* CDS (*Figure 1—figure supplement 3*) and detection of the protein product by immunoblot (*Figure 1—figure supplement 2*). We do also note strong footprint read density of a single amino acid uORF 22 nucleotides from the 5′ end of the ATF4 transcript (*Figure 3E*). Ribosome occupancy of this single methionine is of unclear significance, though it may also play an uncharacterized role in regulating ATF4 translation in myeloma.

We note that in these samples translation elongation was inhibited by cycloheximide. This treatment does not identify uORFs with as high precision as inhibition of initiation complexes by harringtonine or lactimidomycin (*Ingolia et al., 2011*; *Lee et al., 2012*; *Stern-Ginossar et al., 2012*). Thus, we cannot confidently assign 5′ UTR read density across all genes to specific start codons, particularly non-AUG uORFs. However, we did focus on a subset of genes important in the ER stress response. We first looked at *DDIT3* (CHOP) and *PPP1R15A* (GADD34), where in both genes we found strong footprint read density in uORFs known to regulate CDS translation (*Figure 3—figure supplement 2*) (*Jousse et al., 2001*; *Lee et al., 2009*). We next examined other genes with a role in the ER stress response but without known uORF translation. These include *DDIT4* (REDD1), a repressor of mTOR signaling potentially under control of *ATF4* (*Whitney et al., 2009*); *TXNIP*, recently found to be a key mediator of apoptosis under the UPR (*Lerner et al., 2012*; *Oslowski et al., 2012*); and *PPP1R15B*, which also plays a role in the dephosphorylation of eIF2α (*Harding et al., 2009*). Intriguingly, all three genes showed apparent uORF translation, identified as areas of increased 5′ UTR read density with a peak corresponding to AUG or near-AUG initiation codons (*Ingolia et al., 2011*) and bounded by a stop codon on the 3′ end (*Figure 3F–H*, *Figure 3—figure supplement 3*).

## Quantitative proteomics detects only a small subset of anti-apoptotic proteins notably produced

We next explored how the changes at the level of transcription and translation compare to the changes in protein levels during bortezomib-induced apoptosis. At each time point we isolated total protein

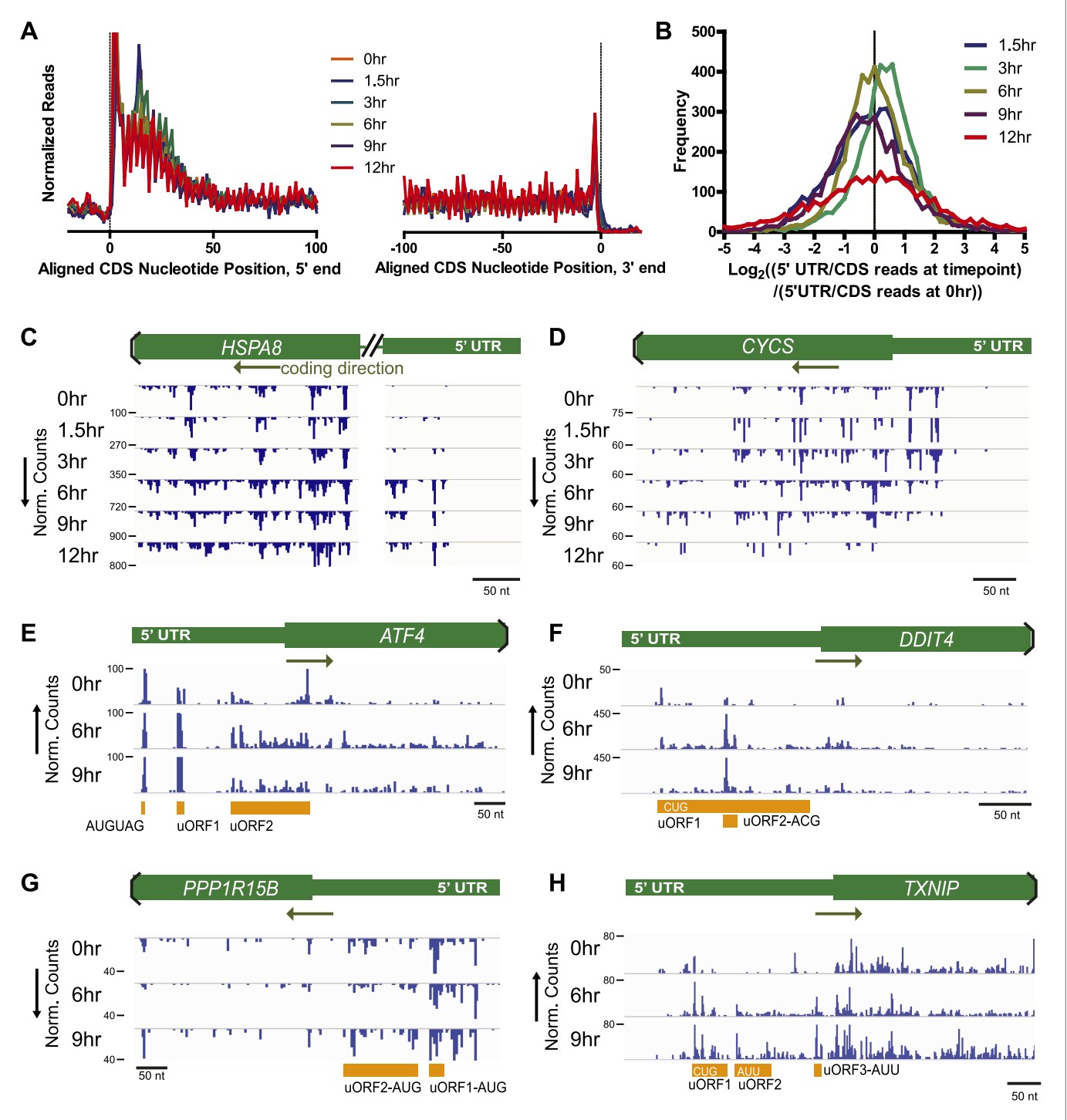

**Figure 3.** Nucleotide resolution of ribosome profiling reveals changes in 5′ UTR translation. (**A**) Metagene analysis, the alignment of footprint reads across all transcripts, to both the 5′ CDS start site and 3′ termination site (*Ingolia et al., 2009*). Reads counts are median-normalized to total aligned reads per sample. No significant changes are noted in overall read alignments during apoptosis. (**B**) Relative translation of 5′ UTR measured by the ratio of 5′ UTR reads to CDS reads per transcript. The log₂ change in this ratio is compared to the 5′ UTR/CDS ratio at 0 hr. (**C** and **D**) Examples of genes with most increased or decreased relative 5′ UTR translation include *HSPA8* (Hsc70 protein) and *CYCS* (cytochrome *c*), respectively. (**E**–**H**) Genes known to be involved in ER stress response show increased read density at 5′ UTR upstream open reading frames (uORFs). *ATF4* shows density at known uORFs as well as a single upstream AUG codon. Putative uORF translation with noted initiation codon is also identified based on areas of increased read density for *DDIT4* (REDD1), *PPP1R15B*, and *TXNIP* (see detailed sequence information in *Figure 3—figure supplement 2*).

*Figure 3. Continued on next page*

*Figure 3. Continued*

The following source data and figure supplements are available for figure 3:

**Source data 1**. 5′ UTR translation.
**Figure supplement 1**. 5′ UTR length.
**Figure supplement 2**. Ribosome profiling demonstrates strong footprint read density in known uORFs of *DDIT3* (CHOP) and *PPP1R15A* (GADD34).
**Figure supplement 3**. Sequence data indicating uORFs suggested by increased footprint read density in *REDD1*, *TXNIP*, and *PPP1R15B* 5′ UTRs.

and employed 6-plex iTRAQ labeling with a different isobaric mass tag (*Mertins et al., 2012*) so that data from all time points can be compared simultaneously in a single mass spectrometry (MS) experiment. After analysis on two different mass spectrometers, a total of 2686 proteins were identified with at least two unique peptides and quantifiable iTRAQ reporter intensities. Typical MS spectra are shown in *Figure 4A*.

We compared the proteins identified by MS as well as those tracked by ribosome profiling to a database of cellular protein abundance (www.pax-db.org; *Figure 4—figure supplement 1*; *Wang et al., 2012*). There was significant overlap between the protein abundance range monitored both by ribosome profiling and iTRAQ proteomics, demonstrating that these two techniques are probing a similar portion of the proteome. We concentrated on 2572 of these proteins that also had corresponding mRNA and ribosome footprint data (*Figure 4—source data 1*). We noted a slight positive correlation ($R = 0.27$) between baseline transcript read density and number of peptides identified by iTRAQ proteomics (*Figure 4—figure supplement 2*). Strikingly, for the vast majority of proteins there is no distinct change in relative protein abundance while large changes are observed in relative mRNA and footprint reads (*Figure 4B*).

Nonetheless, we did find a subset of 12 proteins (0.47% of total) with >50% increase in protein abundance across at least two time points (*Figure 4C*). The majority of these proteins show strong upregulation at the mRNA and footprint level and have potential functions mediating either the folding (*HSPA1A, DNAJB1, BAG3, HSPB1, SERPINH1, CLU*) or degradation (*SQSTM1*) of unfolded proteins. These proteins are key players in an anti-apoptotic response that reduce cellular stress after proteasome inhibition by bortezomib. Interestingly, we also find ferritin light chain (*FTL*) increased (*Figure 4C*). Another group has recently found that depleting ferritin light chain enhances myeloma cell sensitivity to bortezomib (*Campanella et al., 2012*), suggesting it also plays an anti-apoptotic role here. Of note, we did not observe any distinct changes in relative 5′ UTR translation for the few proteins increased by iTRAQ (*Figure 4—figure supplement 3*).

To further validate the iTRAQ findings, we employed an orthogonal method of label-free, targeted, quantitative mass spectrometry termed Selected reaction monitoring (SRM). This method is analogous to Western blotting without the need for dozens of antibodies and potential improvement in performance characteristics (*Maiolica et al., 2012*). We prepared an independent biological time course of MM1.S cells treated with 20 nM bortezomib and isolated tryptic peptides from total cellular lysate. We tracked the relative abundance of one to three peptides from 152 proteins. The SRM peak area intensity data was compared to the iTRAQ reporter signal (e.g., in *Figure 4A–D*, respectively; heat map in *Figure 4—figure supplement 4*). Comparing across all time points in *Figure 4E* we found a strong correlation between SRM and iTRAQ data ($R = 0.80$). For the few increased proteins the abundance changes detected by SRM are greater than those found by iTRAQ, which is consistent with the greater suppression of iTRAQ reporter ions often seen in complex samples (*Ow et al., 2009*). The label-free SRM data do not suffer from this same limitation and more likely reflect the true change in protein abundance. This independent MS assay confirms that only a small subset of proteins show measurable increases in abundance during apoptosis.

## Global translational shutdown during apoptosis

Our quantitative proteomic results are strongly consistent with polysome analysis (*Figure 5—figure supplement 1*) showing a significant decrease in translation well before any loss in cell viability. We wished to better understand the mechanism leading to this translational shutdown during apoptosis.

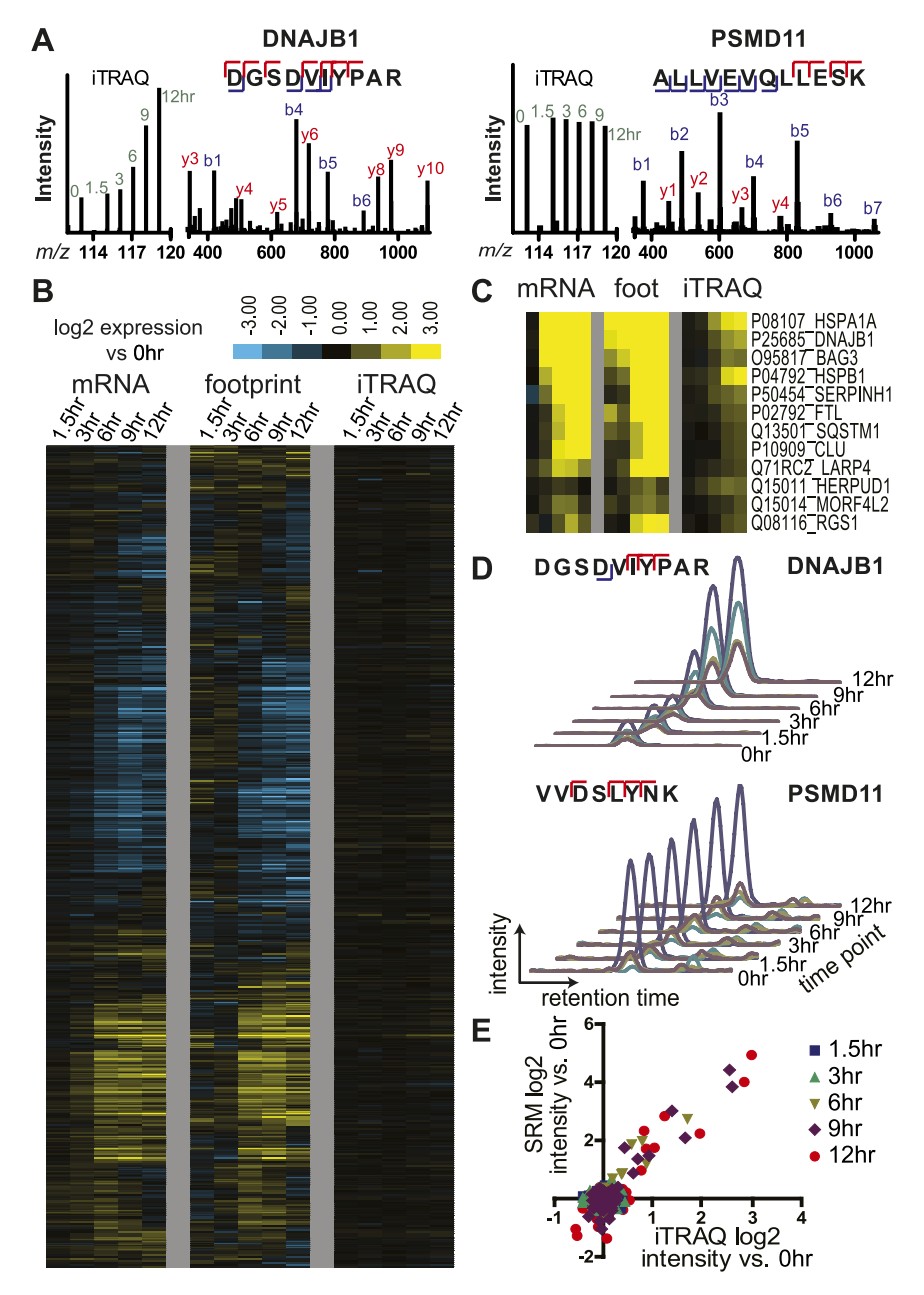

**Figure 4**. Unbiased and targeted quantitative proteomics reveals abundance changes in a small subset of proteins. (**A**) Example mass spectra demonstrating both *m/z* peaks used for peptide sequencing and iTRAQ reporter ion signal to measure relative abundance across time points. (**B**) Hierarchical clustering heat map of paired mRNA and ribosome footprint relative read density vs 0 hr with relative iTRAQ protein abundance for 2572 proteins. (**C**) Inset of heat map for proteins increased in relative abundance by >50% at ≥2 time points shows few proteins are measurably produced during bortezomib-induced apoptosis. (**D**) Targeted selected reaction monitoring (SRM) assays orthogonally validate iTRAQ data for 152 proteins. In this representative data, each colored trace monitors the intensity of a given parent and fragment ion pair, as demarcated in the peptide sequence; multiple co-eluting peaks positively identify a targeted peptide. (**E**) Protein abundance measured by both SRM and iTRAQ demonstrate strong correlation across time points (*r* = 0.80).

The following source data and figure supplements are available for figure 4:

**Source data 1**. Proteomic data.

*Figure 4. Continued on next page*

*Figure 4. Continued*

**Figure supplement 1**. Protein abundance comparison.

**Figure supplement 2**. Comparison of baseline (0 hr) read density of mRNA transcripts vs number of identified peptides mapping to each protein in iTRAQ proteomics.

**Figure supplement 3**. Relative 5′ UTR translation across the time course for upregulated genes.

**Figure supplement 4**. Unbiased and targeted proteomics comparison to deep sequencing data.

During the UPR, it is well-known that phosphorylation of the initiation factor eIF2α by the ER-resident sensor kinase PERK is associated with general translation inhibition (*Walter and Ron, 2011*). At early time points we find that PERK appears phosphorylated, as previously reported (*Atkins et al., 2013*), and at later time points it undergoes proteolysis (*Figure 5A*). In contrast to other drugs that induce ER stress, we find levels of eIF2α phosphorylation actually decrease after bortezomib exposure (*Figure 5A*). This finding suggests that there must be other drivers of translation inhibition in this system.

Phosphorylation and dephosphorylation of proteins in the eIF4 complex may also inhibit mRNA cap-dependent translation, including hypophosphorylation of 4E-BP1 (*Spriggs et al., 2010*). Indeed, we observed 4E-BP1 dephosphorylation at the 6 hr time point. This is followed by rapid caspase degradation of 4E-BP1 during cell death (*Figure 5A*). It is also known that caspases target other proteins in the eIF4 complex (*Bushell et al., 2004*). Using N-terminomics (described in more detail below) we found that proteolysis of initiation factors begins at the 6 hr time point and accelerates later in apoptosis (*Figure 6—figure supplement 1*). Another potentially important mechanism is the global degradation of mRNA during apoptosis (*Del Prete et al., 2002*) (*Figure 1B*), depriving ribosomes of substrates for translation. Together, these results demonstrate that well before the loss of cellular viability, translation is inhibited both by shutting down translation initiation and destruction of mRNA.

## A quantitative mass-action model describes protein changes

We sought a quantitative explanation for the limited changes in relative protein level we observed despite large changes in relative transcript and ribosome footprints. Recently developed systems-level technologies, as we use here, enable genome-wide quantitative assessment of the efficiency for decoding of mRNA to protein (*Vogel and Marcotte, 2012*). In this apoptotic system, proteins are relatively stable due to several factors: at 20 nM bortezomib in MM1.S cells, proteasomal activity is almost completely inhibited (*Berkers et al., 2005*). The proteasome is also extensively cut by caspases leading to loss of activity (*Gray et al., 2010*; *Figure 6—figure supplement 1*). Moreover, we find that endoproteolysis during apoptosis cuts fewer than 20% of the proteins in the cell and often only once or twice per protein leading to stable domains (*Dix et al., 2008*; *Mahrus et al., 2008*; *Crawford et al., 2013*). These domains would be indistinguishable from intact proteins by iTRAQ. Therefore, we expect that detectable protein degradation is extremely limited and changes by iTRAQ largely reflect protein production alone. In contrast, mRNAs with very different stabilities (steady-state half-lives on the order of minutes to hours) are rapidly degraded with similar kinetics under various apoptotic inducers even prior to cell death (*Del Prete et al., 2002*).

Mass-action models, as shown in *Figure 5B*, can describe the dynamic relationship between protein and mRNA but require absolute abundance measurements per cell. As described in (*Schwanhausser et al., 2011*), we used measurements of total mRNA combined with mRNA-seq read density to estimate the absolute abundance of each transcript per cell in untreated MM1.S cells. For absolute protein abundance we used a recently described method of label-free quantitation termed intensity based absolute quantitation (iBAQ) (*Schwanhausser et al., 2011*). From untreated MM1.S cells we generated absolute abundance estimates for 3369 proteins (*Figure 5—source data 1*). Compared to baseline protein abundance, we found a slightly stronger correlation for footprint read density than mRNA (*Figure 5—figure supplement 2*). Interestingly, we also found that increased 5′ UTR translation at baseline may correlate with decreased steady-state protein abundance (*Figure 5—figure supplement 2*). Overall distributions of protein and mRNA copies per MM1.S cell, with a median of 11 transcript copies and ~16,000 protein copies (*Figure 5C*), were similar to those previously found in mouse NIH3T3 cells (*Schwanhausser et al., 2011*).

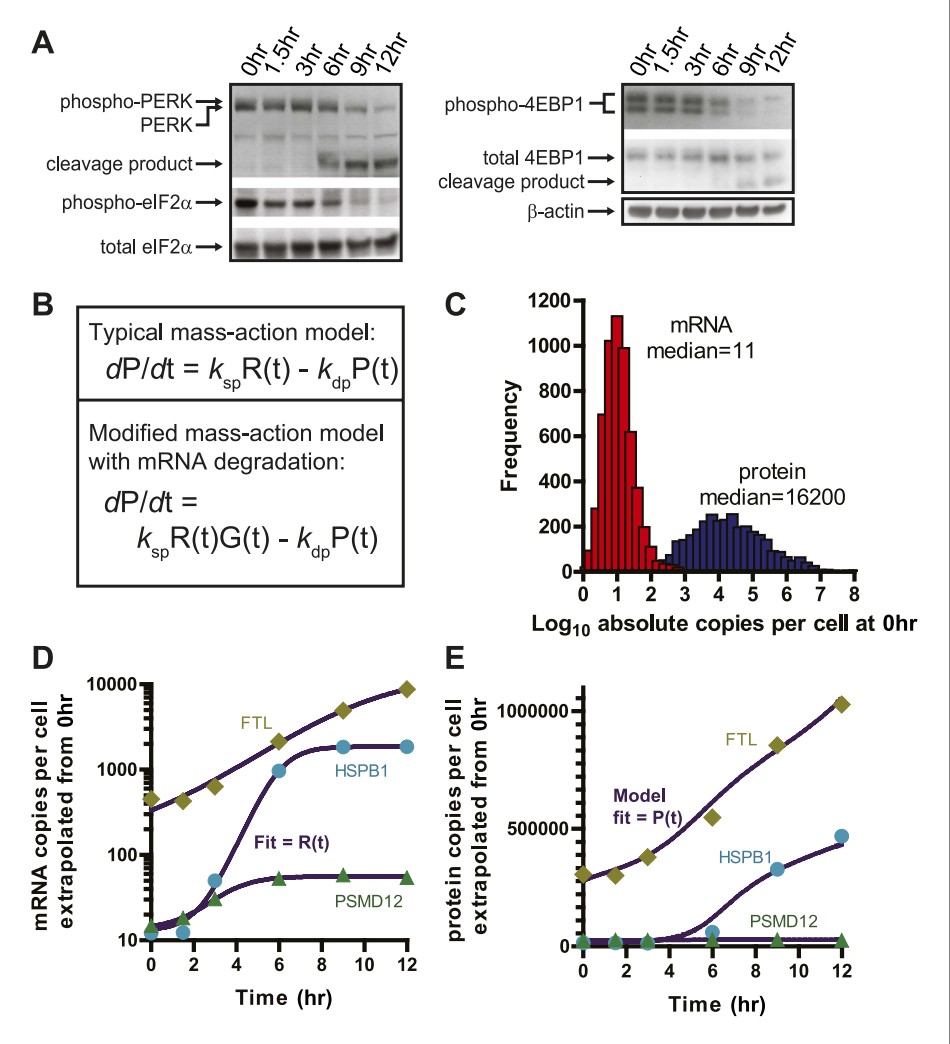

**Figure 5**. Translational shutdown during apoptosis and quantitative modeling reconciles deep sequencing and proteomic data. (**A**) Western blotting tracked UPR proteins important in modulating translation via ER stress. We detect the known caspase cleavage of PERK and 4E-BP1. (**B**) Differential rate equations describing protein production. R(t) = mRNA abundance; P(t) = protein abundance; $k_{sp}$ = protein translation rate constant; $k_{dp}$ = protein degradation rate constant. G(t) describes global mRNA degradation (***Figure 1B***). (**C**) Absolute copies per cell at 0 hr of both mRNA (~5700 transcripts) and protein measured by iBAQ (~3400 proteins) demonstrates distributions of same order of magnitude as those seen in mouse NIH3T3 cells (***Schwanhausser et al., 2011***). (**D**) Extrapolation of absolute mRNA copy numbers per cell based on RPKM, not accounting for mRNA degradation, is well-fit by either a sigmoid or quadratic function to define R(t). (**E**) Absolute protein copies per cell were extrapolated from iBAQ results based on SRM assay intensity. The modified mass-action model incorporating mRNA degradation was able to well-fit the subset of proteins that were detectably increased and also explained why others increased at the transcriptional level did not show protein increases.

The following source data and figure supplements are available for figure 5:

**Source data 1**. mRNA and protein absolute abundance.

**Figure supplement 1**. Polysome profiles.

**Figure supplement 2**. Comparison between deep sequencing and proteomic data.

**Figure supplement 3**. mRNA-seq data scaled by global mRNA degradation.

*Figure 5. Continued on next page*

*Figure 5. Continued*

**Figure supplement 4**. Model fits to 13 selected genes.

**Figure supplement 5**. Predicted protein changes from mRNA changes.

We used these absolute abundance measurements to prepare an approximate quantitative model in our system. We built a mass-action model similar to that in (*Schwanhausser et al., 2011*) but modified to incorporate a term to account for global mRNA degradation (*Figures 1B and 5B*) (see 'Materials and methods' for details of modeling). This model describes the change in protein copies per cell as a function of transcript copies per cell, the rate of translation, and the rate of protein degradation. We analyzed a subset of 13 genes with transcriptional increase but non-uniform changes in protein levels (*Figure 5D*, *Figure 5—figure supplement 4*). This model was able to fit the protein abundance data, primarily by varying the term $k_{sp}$, describing the number of proteins produced per transcript per hour (*Figure 5E*, *Figure 5—figure supplement 4*). $k_{sp}$ ranged from 10 to 270, consistent with the range found previously (*Schwanhausser et al., 2011*).

Based on this model, we can begin to reconcile our deep sequencing and proteomic data. During the early stages of apoptosis, the group of transcripts with multi-fold increases in relative abundance do not all lead to measurable protein increases. This is because the absolute number of these transcripts per cell in the sample remains fairly stable as transcription and mRNA degradation offset. This is illustrated by scaling the relative mRNA data shown in *Figure 4A* by a factor reflecting global mRNA degradation (*Figure 5—figure supplement 3*). Notably, the few proteins that are increased tend to demonstrate increases in absolute transcript level at later time points. Our model further shows that with the same fold-increase in transcript level, proteins with low baseline abundance will have more dramatic increases in relative protein concentration when compared to proteins with high baseline abundance (*Figure 5—figure supplement 5*).

## Cellular struggle with chemotherapy does not affect apoptotic proteolysis

It is unknown how the cellular response to chemotherapeutic treatment influences cellular deconstruction by proteases during apoptosis, including both caspase and non-caspase proteases (*Pop and Salvesen, 2009*; *Moffitt et al., 2010*). Our laboratory has developed an N-terminomics technology using the engineered enzyme subtiligase to specifically label and enrich for free protein N-termini generated by proteolysis in complex biological samples (*Mahrus et al., 2008*). Others have also identified caspase cleavage substrates after treatment with chemotherapeutics with different N-terminomics approaches (*Gausdal et al., 2008*; *Impens et al., 2008*). By MS analysis in a variety of cell lines under both apoptotic and non-apoptotic conditions our laboratory has identified >8000 protein cleavage sites, including >1700 putative caspase substrates with an Asp immediately N-terminal to cleavage site (P1 position). We have compiled these cleavage sites into the 'DegraBase' (*Crawford et al., 2013*); wellslab.ucsf.edu/degrabase).

We have previously found that different apoptotic inducers result in different patterns of caspase substrate cleavage, though the reasons for these differences remain unknown (*Shimbo et al., 2012*). We can now directly compare whether the cellular response to chemotherapy at the transcriptional and translational level influences the cleavage of substrates during terminal apoptosis. For each Cluster (*Figure 1C*) we queried the DegraBase to investigate if a potential substrate was identified in the DegraBase, if it was cut by caspases, and, if so, the relative frequency of caspase cleavage. If the cellular response to bortezomib induced common anti-apoptotic pathways targeted by the caspases, it was possible that genes in Cluster Upreg would be more prominently featured as substrates in the DegraBase. However, we found no evidence that genes in any Cluster were significantly enriched (or de-enriched) among identified caspase substrates (*Figure 6A*).

These findings suggest that protease substrates are not targeted based on changes in their abundance. We further explored this hypothesis by monitoring the kinetic appearance of proteolytic substrates with a quantitative SRM assay (*Agard et al., 2012*; *Shimbo et al., 2012*). We included both non-caspase and caspase-cleaved peptides in our SRM assay with a preference for those found in multiple experiments under apoptotic conditions. Interestingly, many of the chaperone and co-chaperone proteins, relatively increased at the transcriptional and translational levels, had only

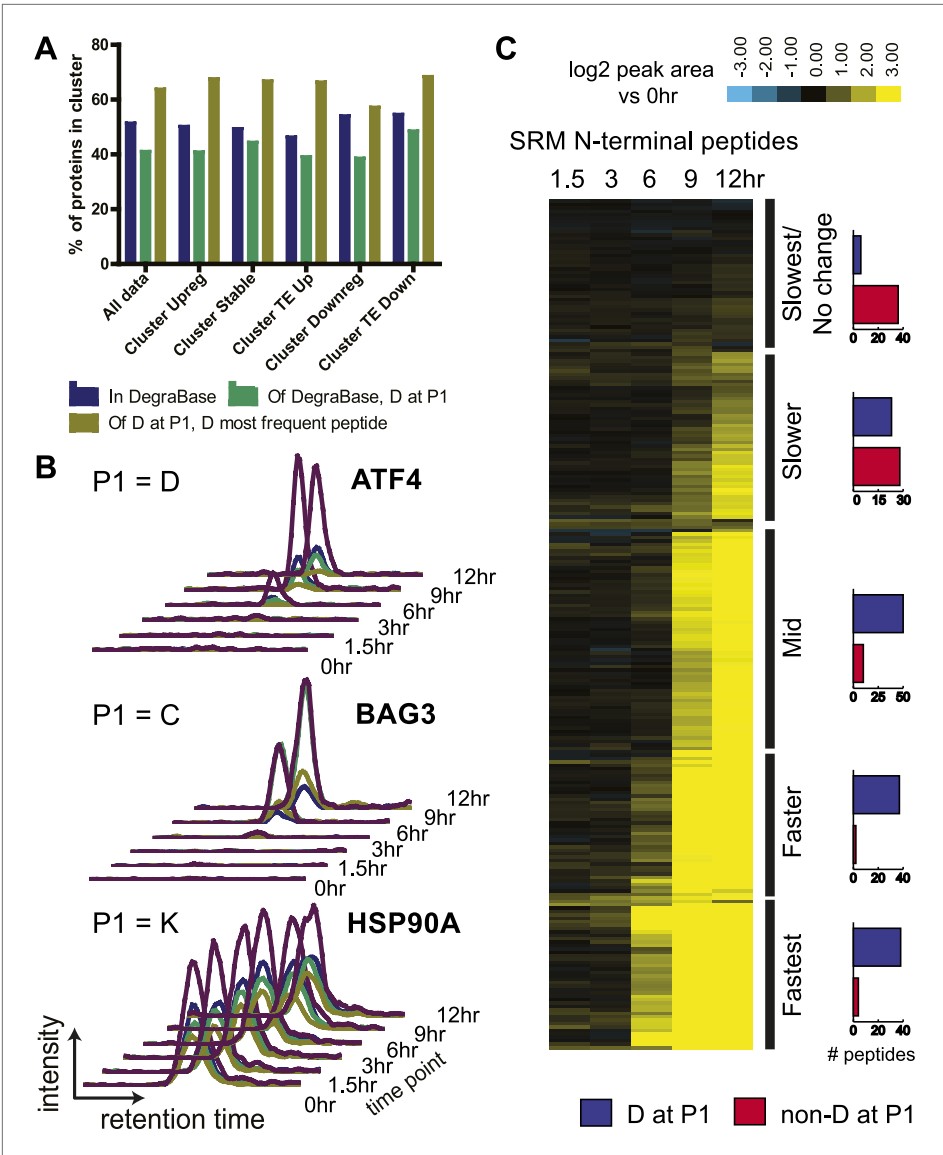

**Figure 6**. N-terminomics tracks proteolytic cleavage substrates. (**A**) We compared defined hierarchical clusters (*Figure 1C*) to all proteolytic cleavage substrates present in the DegraBase, a database of over 8000 cleavage events (*Crawford et al., 2013*). D at P1 suggests a putative caspase substrate. We find no significant variation in levels of proteolysis by $\chi^2$ analysis of each cluster compared to the set of all substrates. (**B**) Representative SRM intensity for appearance of proteolytically-processed peptides, both caspase- and non-caspase-cleaved (*Figure 6—source data 1*). Each trace represents a single parent ion/fragment ion transition; co-elution of multiple targeted transitions confirms peptide. (**C**) Heat map showing relative SRM intensity of 252 proteolytic peptides monitored in this system compared to 0 hr baseline. Five groups defined by unsupervised hierarchical clustering representing relative cleavage kinetics show caspase proteolysis generally precedes other apoptotic proteolytic events.

The following source data and figure supplements are available for figure 6:

**Source data 1**. Degradomics SRM assay.

**Figure supplement 1**. N-terminomics biological data subgroups.

**Figure supplement 2**. N-terminomics data with caspase and non-caspase cleavage sites in same substrate.

non-caspase proteolytic peptides identified in the DegraBase. These included Hsp70-1, Hsp40, Hsp27, BiP (*HSPA5*), Hsc70 (*HSPA8*), and BAG3. The highly abundant and constitutively expressed chaperone Hsp90-β had numerous non-caspase cleavages and a single caspase-cleaved peptide.

We used this SRM assay to explore whether proteins upregulated in response to bortezomib were more rapidly cleaved during apoptotic proteolysis. In *Figure 6B* we show example SRM intensity data while *Figure 6C* displays a heat map of similar SRM intensity data compared to the untreated baseline across all 252 peptides targeted. Caspase-cleaved peptides begin to appear at the 6 hr time point, when caspase activity was also detected biochemically (*Figure 1B*). In *Figure 6—figure supplement 1* we subdivide the monitored peptides based on biological function. We do not find that stress response proteins are cleaved more rapidly than those in other groups. However, we note rapid cleavage of ATF4 (as seen before, [*Shimbo et al., 2012*]), BAG3, and Hsp90-β. It is possible that these particular proteins represent important nodes in deconstructing the cellular stress response after borte-zomib. Generally, our results suggest that drug-specific response at the transcriptional and translational level does not broadly affect protease dynamics once apoptosis ensues.

## Non-caspase proteolytic events with caspase-like kinetics

Our prior studies examining quantitative appearance of proteolytic substrates only focused on caspase substrates (*Agard et al., 2012*; *Shimbo et al., 2012*). Here, we also monitored the kinetics of appearance of non-caspase proteolytic events during apoptosis. We tracked 163 caspase-cleaved peptides (D at P1) and 89 peptides derived from non-caspase proteolysis. The list includes 52 proteins that were cleaved by both caspase- and non-caspase proteases to examine the relative kinetics of different proteolytic events in the same substrate (*Figure 6—figure supplement 2*).

Unsupervised hierarchical clustering was used to rank the rates of proteolytic cleavage events (*Figure 6C*). We defined five broad groups ranging from the Fastest cleavages, with robust substrate appearance at 6 hr, to Slowest/No Change, with no increase in cleavage even at late apoptosis. Interestingly, the large majority of non-caspase-cleaved substrates appear in the Slow and Slowest/No Change groups. While this only represents a small portion of the >3,000 non-caspase endoproteolytic cleavages in the DegraBase, this finding suggests that most non-caspase proteolytic events in apoptotic cells are a consequence of and not a driver of apoptosis itself.

However, we did find 16 non-caspase-cleaved peptides appearing in either the Fastest, Faster, or Mid categories (*Table 2*). Of the 89 non-caspase peptides monitored, 24 were tryptic-like (K or R at P1). Yet none of the 16 most rapidly cleaved peptides had a tryptic-like cleavage and only one has been previously annotated in MEROPS, the largest current database of proteolytic cleavage (*Rawlings et al., 2012*). These 16 peptides were cleaved with kinetics similar to caspases, suggesting that they may play roles in driving apoptosis. Among these we noted a relative preponderance of Pro (3 peptides), Gln (4 peptides), and Glu (4 peptides) residues at the P1 site (*Table 2*). While caspases may occasionally cleave at Glu (*Krippner-Heidenreich et al., 2001*), none of the other classic proteases thought to be active during MM1.S cell apoptosis, including the cathepsins or calpains, have a preference for Pro, Gln, or Glu at the P1 site (as described in MEROPS). Our findings suggest that, at least in a limited set of substrates, non-Asp cleavages may be mediated by uncharacterized proteases acting with similar kinetics as the caspases during apoptosis.

## An integrated approach informs combination therapeutic regimens

Genes upregulated at both the transcriptional, translational, and protein level may represent the cellular protective response to avoid apoptosis. We thus hypothesized that by inhibiting these 'first responders' to the chemotherapeutic treatment we may be able to drive the malignant cell toward apoptosis and reduce therapeutic resistance. To this end we identified the most prominent upstream regulators of the genes in Cluster Upreg (*Table 1*). These include NRF2, a transcription factor regulating the hypoxic response; HSF1, a transcription factor regulating the heat shock response; and PERK, the ER-resident kinase. HSF1 in particular regulates the transcription of many genes we also found increased by proteomics experiments (*Mendillo et al., 2012*).

We therefore explored whether HSF1 inhibition would enhance cell death by bortezomib in MM1.S cells. We used a recently described small molecule inhibitor of HSF1, KRIBB11 (*Yoon et al., 2011*). This drug was added with and without a low dose of 2.5 nM bortezomib, which alone led to ~50% cell death over 48 hr. At the 24 hr time point, the combination of KRIBB11 and bortezomib caused significant cell death, while neither compound alone decreased cell viability (*Figure 7A*).

**Table 2.** Comparison of caspase- and non-caspase cleavages during apoptotic time course as monitored by N-terminomics and SRM

| | |
|---|---|
| Total D = P1 peptides | 163 |
| Total non-D = P1 peptides | 89 |

| | |
|---|---|
| Tryptic-like (K or R at P1) | 24 |
| Non-tryptic (all other residues at P1) | 65 |

| Cluster | D at P1 | Non-D at P1 |
|---|---|---|
| *Slowest/no change* | 6 | 36 |
| *Slow* | 23 | 28 |
| *Mid* | 50 | 10 |
| *Faster* | 37 | 2 |
| *Fastest* | 38 | 4 |

| **Non-D peptides in fastest three clusters** | | **UniProt** | **Gene** | **P1** | **Sequence Position** | **MEROPS annotated?** |
|---|---|---|---|---|---|---|
| *Fastest* | (C)GQVAAAAAAQPPASHGPER | O95817 | BAG3 | C | 151 | N |
| | (S)AVGFNEMEAPTTAYK | P14317 | HCLS1 | S | 208 | N |
| | (P)GHGSGWAETPR | O75533 | SF3B1 | P | 304 | N |
| | (Q)VLTVPATDIAEETVISEEPPAKR | Q06547 | GABP1 | Q | 306 | N |
| *Fast* | (Q)ALKEEPQTVPEMPGETPPLSPIDMESQER | P05412 | JUN | Q | 223 | N |
| | (P)AVNGATGHSSSLDAR | Q07817 | BCL2L1 | P | 63 | N |
| *Mid* | (E)AAGATGDAIEPAPPSQGAEAK | P49006 | MARCKSL1 | E | 56 | N |
| | (Q)AASGDVQTYQIR | P16220 | CREB1 | Q | 243 | N |
| | (E)GGIDMDAFQER | P29083 | GTF2E1 | E | 297 | N |
| | (A)SIFGGAKPVDTAAR | P23588 | EIF4B | A | 358 | Y–meprin alpha (*Becker-Pauly et al., 2011*) |
| | (E)AIQNFSFR | O75122 | CLASP2 | E | 946 | N |
| | (P)HFEPVVPLPDKIEVK | P49792 | RANBP2 | P | 1170 | N |
| | (N)SWFENAEEDLTDPVR | Q13813 | SPTAN1 | N | 2104 | N |
| | (L)AFSEQEEHELPVLSR | O75995 | SASH3 | L | 128 | N |
| | (Q)AIMEMGAVAADKGKK | O95817 | BAG3 | Q | 522 | N |
| | (E)AILEDEQTQR | Q9P2E9 | RRBP1 | E | 1298 | N |

By Fisher's exact test there is a significant absence of tryptic-like cleavages in the 16 rapidly cleaved peptides (p=0.005).

There are no known small molecules which specifically inhibit NRF2, but we could test whether PERK inhibition could sensitize cells to death using a newly described small molecule PERK inhibitor, GSK2606414 (*Axten et al., 2012*). Indeed, PERK inhibition did sensitize cells to death by bortezomib (*Figure 7B*). For both the HSF1 and PERK inhibitors the most robust effects were seen at concentrations in excess of the in vitro $IC_{50}$ (~1.2 µM for KRIBB11, ~1 nM for GSK2606414). Notwithstanding potential off-target effects, these results would support the notion that integrated systems-level data can inform potential chemotherapeutic combinations.

## Discussion

Here we present a systems-level study of how malignant cells respond to apoptosis induced by a clinically relevant chemotherapeutic. Our results show how myeloma cells initially struggle to adapt to the proteasome inhibitor bortezomib. A general inhibition of translation coupled with mRNA degradation leads to surprisingly little change in overall protein levels despite relative transcriptional increases in many genes. Furthermore, global protein levels are maintained due to inhibition of proteasomal activity by

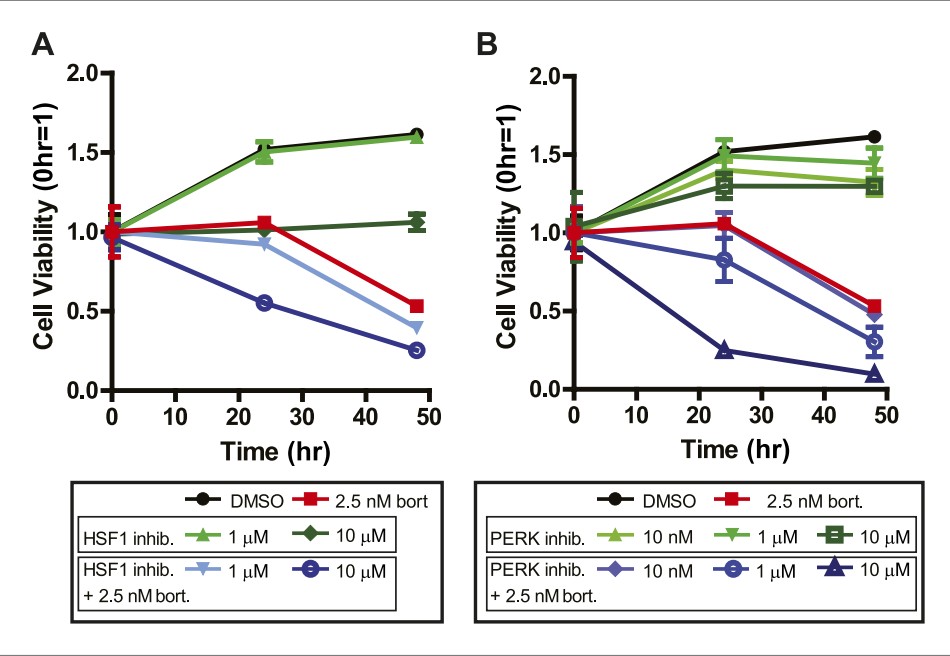

**Figure 7**. Systems-level analysis suggests potential chemotherapeutic combinations with bortezomib in myeloma.
(**A**) Bort = bortezomib. HSF1 inhibitor used was KRIBB11 (*Yoon et al., 2011*) and shows additive effect with bortezomib.
(**B**) PERK inhibitor was GSK2606414 and shows additive effect with bortezomib (*Axten et al., 2012*). All measurements
in 24-well plates in triplicate; mean ± SD shown.

bortezomib and extensive cleavage of proteasomal subunits by caspases (*Gray et al., 2010*). The cell
also modulates translational control of stress responsive proteins related to bortezomib treatment.
Co-targeting key regulators of these processes such as HSF1 and PERK can further accelerate apoptosis. Myeloma cells ultimately succumb to the pharmacologic insult, leading to activation of the caspases as well as one or more non-caspase proteases that have yet to be characterized. While pieces of
this story have been told by other experiments, the approach we present here, examining the integrated regulation of the transcriptional, translational, and proteolytic machinery, gives a global and
in-depth view of the apoptotic process.

Translational regulation during apoptosis has only been examined in a limited context. Bushell and
coworkers examined the relative changes in translational efficiency during early TRAIL-induced apoptosis by polysome profiling (microarray analysis of mRNA associated with translating ribosomes)
(*Bushell et al., 2006*). Genes with the greatest degree of decrease in TE included many translation-related proteins, similar to our findings. Those with the largest increase, however, did not overlap with
our results, suggesting that positive changes in TE may be specific to the apoptotic inducer or induction of the intrinsic vs extrinsic apoptotic pathways.

Importantly, after proteasomal inhibition in myeloma cells we found that translational reprogramming favors proteins which help the cell adapt to unfolded protein stress. Possible mechanisms for the
alteration of translation efficiency involve preferential translation via internal ribosome entry sites
(IRES) (*Bushell et al., 2006*), partitioning of transcripts toward ER-bound ribosomes under stress (*Reid
and Nicchitta, 2012*), or other uncharacterized regulatory programs governing ribosomal-transcript
association. We also identified uORF translation in genes related to the ER stress response, *TXNIP*,
*PPP1R15B*, and *DDIT3* (REDD1), that had not previously been characterized. The exact interplay of
uORF and CDS translation for these genes remains to be investigated, but it is possible there is a
regulatory function similar to that already known for ATF4, CHOP, and GADD34. We also found that
differential 5′ UTR translation appears common in the cellular response to bortezomib but relates
variably to increased CDS translation. It will be interesting to compare the effects on translation
regulation in different cell types exposed to chemotherapeutics with different mechanisms of action to
find whether these changes are unique to bortezomib. Incorporation of translation initiation inhibitors

in ribosome profiling and paired-end reads in mRNA-seq can also provide novel insight into translation regulation through generation of custom proteomic databases (*Stern-Ginossar et al., 2012*; *Menschaert et al., 2013*; *Sheynkman et al., 2013*).

Surprisingly, we found that eIF2α phosphorylation did not play a major role in translation inhibition in this apoptotic system involving the UPR. Others have also observed decreased eIF2α phosphorylation in MM1.S cells after proteasome inhibition (*Parlati et al., 2009*). However, eIF2α phosphorylation is thought to be necessary for increased ATF4 translation (*Lu et al., 2004*). The mechanism leading to the induction of ATF4 in this system (*Figure 1—figure supplement 2*) is therefore unclear, but may relate to translation of the additional uORF we identified (*Figure 3E*).

Our paired transcriptomic and proteomic data stands in stark contrast to studies of heat and osmolar shock in yeast, where increases at the transcript level are well-correlated with increases in protein (*Lee et al., 2011*; *Lackner et al., 2012*). We used a quantitative model incorporating global mRNA degradation to explain this contrast with non-apoptotic systems. While our proteomic analysis is quite extensive, this mass-action model suggests that low abundance proteins not detected by MS, such as CHOP or XBP1 (*Figure 1—figure supplement 2*), are more likely to be increased in relative protein abundance after transcriptional changes. Future advances in proteomic techniques may allow us to detect more biologically relevant protein changes. The mass-action model also suggests the kinetics of protein production from mRNA transcripts largely governs protein abundance (*Schwanhausser et al., 2011*). In a system undergoing less rapid apoptosis, such as those explored at the single cell level (*Cohen et al., 2008*; *Geva-Zatorsky et al., 2010*), there may be greater opportunity for proteins to accumulate and reveal relationships between ribosome footprint occupancy and protein production.

Previous work showed that caspase substrates were largely conserved at the identification level after different chemotherapeutic treatments, though different quantitative signatures of caspase proteolysis could be observed (*Shimbo et al., 2012*). Here we find that in the setting of rapid apoptosis, proteins with changes at the transcriptional and translational level do not appear to be targeted more efficiently by proteases during apoptosis. Under conditions of more prolonged apoptosis, however, changes in the proteome may influence the appearance of proteolytic substrates and this remains to be investigated. Furthermore, our identification of non-caspase proteolytic events occurring with similar kinetics to caspase cleavage suggests other proteases are being activated during apoptosis. The biological role of these proteases during apoptosis awaits exploration.

Based on our systems-level analyses, we identified potential small molecules which may be effective in combination with bortezomib in myeloma therapy. Inhibition of HSF1 and PERK have also recently been proposed by others as promising therapeutic targets in myeloma (*Heimberger et al., 2012*; *Atkins et al., 2013*) and represents a proof of principle for our approach. Our deep sequencing and proteomic results suggest in particular that the combination of bortezomib with HSF1 inhibition could be beneficial in myeloma therapy.

We hypothesize that using a similar systems-based approach with less well-studied therapeutics will reveal novel means by which cells attempt to evade apoptosis, not just at the transcriptional level but also at the translational and post-translational levels. This analysis will likely complement information gained by genome-wide RNA knockdown approaches (*Zhu et al., 2011*). By targeting these early response pathways identified at multiple systems levels cancer cell death can be maximized and resistance avoided. Furthermore, changes in specific proteins revealed by these analyses may be monitored after treatment to establish novel biomarkers of chemotherapeutic efficacy or drug resistance. We anticipate that integrated, systems-level approaches, such as that presented here, will help to form the future basis of evaluating preclinical chemotherapeutic effects.

## Materials and methods

### Cell culture and drug treatment

MM1.S cells (obtained from ATCC, Manassas, VA, USA) were grown in suspension to $1 \times 10^6$ cells/ml in RPMI-1640 media with 10% FBS. Bortezomib (LC Laboratories, Woburn, MA, USA) 20 µM stock solution in sterile-filtered phosphate buffered saline (PBS) was simultaneously added to a final concentration of 20 nM to flasks each containing $300 \times 10^6$ cells (PBS only added to control sample). At the indicated time point cells were separated into aliquots for each experimental approach ($15 \times 10^6$ cells

in duplicate for each of mRNAseq, iTRAQ proteomics, and ribosome footprinting; 200 × 10⁶ cells for N-terminomics). Cells for ribosome footprinting alone were incubated at 37°C for 1 min with 100 µg/mL cycloheximide (Sigma-Aldrich, St. Louis, MO, USA) from 50 mg/ml stock in 100% EtOH. All cells were washed in PBS (PBS + 100 µg/ml cycloheximide for ribosome footprint samples), pelleted by centrifugation, and flash frozen in liquid $N_2$, then stored at −80°C. Cell viability and caspase activity were assessed by Cell-Titer Glo and Caspase-Glo (Promega, Madison, WI, USA) assays per manufacturer protocol, respectively. KRIBB11 and GSK2606414 were purchased from EMD Millipore (Billerica, MA, USA) and 10 mM stock solutions prepared in DMSO and stored in single-use aliquots at −80°C. These drugs and bortezomib were added as indicated in triplicate in 24-well plates at an MM1.S cell density of 0.5 × 10⁶ cells/ml.

## Measurement of total RNA and mRNA

Duplicate samples of 4 × 10⁶ MM1.S cells at each time point after 20 nM bortezomib treatment were pelleted at 800×*g* for 5 min, supernatant aspirated, resuspended in ice-cold PBS, pelleted again, and supernatant aspirated. Total RNA was extracted using Trizol reagent (Invitrogen, Carlsbad, CA, USA) following manufacturer's protocol; lysis was performed by passing 20x through a 22½ gauge needle. Total RNA was resuspended in 100 µl RNAse-free water and further purified using RNAeasy kit (Qiagen, Germantown, MD) with on-column DNAse I treatment per manufacturer's protocol. Purified RNA pellet was resuspended in 100 µl RNAse-free water. Total RNA concentration (as shown in *Figure 1B*) was measured spectrophotometrically using a NanoDrop ND-1000 UV-Vis spectrophotometer (Thermo Fisher, Waltham, MA, USA).

mRNA was further purified by poly(A) separation using Oligo (dT)$_{25}$ Magnetic Beads kit (New England BioLabs, Ipswich, MA, USA). Total RNA samples were diluted with 500 µl of kit Lysis/Binding buffer and bound to of equilibrated beads (100 µl bead slurry used). Beads were washed once in 500 µl Wash Buffer one followed by one wash in 500 µl Wash Buffer two. mRNA was eluted in 40 µl Elution Buffer and concentration measured by NanoDrop.

## Immunoblotting and protein measurements

A separate time course of MM1.S cells was prepared and treated with 20 nM bortezomib as described above 10 × 10⁶ MM1.S cells harvested at indicated time points were lysed in RIPA buffer (50 mM Tris-HCl, 150 mM NaCl, 1% NP-40, 0.5% sodium deoxycholate, 0.1% SDS, pH 8.0) supplemented with 5 mM EDTA and 1x protease inhibitor cocktail (Cell Signaling Technology, Danvers, MA, USA). Protein concentration from cell lysates was measured by BCA assay (Thermo Fisher, Waltham, MA, USA). ~25 µg of total protein at each time point was separated on NuPAGE 4–12% Bis-Tris polyacrylmide gels (Invitrogen) and transferred to 0.45 µM PVDF membrane. Primary antibodies used for immunoblotting were obtained from Cell Signaling Technology (α-CHOP, α-XIAP, α-Bid, α-PERK, α-4E-BP1, α-phospho(Thr37/46)-4E-BP1, α-eIF2α, α-phospho(Ser51)-eIF2α), Santa Cruz Biotechnology (Santa Cruz, CA, USA) (α-CREB2 [ATF4]), BioLegend (San Diego, CA, USA) (α-XBP1s), and Sigma (α-β-actin).

## Ribosome profiling and mRNAseq sample preparation and library generation

Harvested cell pellets for ribosome profiling were suspended and lysed in 500 µl ice-cold polysome lysis buffer (20 mM Tris, pH 7.4, 250 mM NaCl, 15 mM MgCl₂, 1 mM dithiothreitol, 0.5% Triton X-100, 24 U/ml Turbo DNase (Ambion, Austin, TX, USA), and 100 µg/ml cycloheximide) by repeated pipetting. Lysate was clarified by centrifugation for 10 min at 20,000×*g* at 4°C. 3 µl of RNase I 100 U/µl (Ambion) was added and sample incubated for 45 min at room temperature. The digestion was stopped by the addition of 10 µl SuperaseIn 20 U/µl (Ambion). Digested samples was then loaded onto a 1 M sucrose cushion, prepared in polysome buffer plus 0.1 U/µl SuperaseIn. Ribosomes were pelleted by centrifugation for 4 hr at 70,000 rpm, 4°C in a TLA-110 rotor. The liquid was removed and the pellet was resuspended in 600 µl 10 mM Tris (pH 7), followed by the immediate addition of 40 µl 20% SDS. The sample was heated to 65°C and RNA was extracted using two rounds of acid phenol/chloroform followed by chloroform alone. RNA was precipitated from the aqueous phase by adding sodium acetate to a final concentration of 300 mM followed by at least one volume of isopropanol. Precipitation was carried out at −20°C overnight and RNA was then pelleted by centrifugation for 30 min at 20,000×*g*, 4°C. The supernatant was discarded, the pellet was air-dried, and the RNA was resuspended in 25 µl Tris (pH 7). RNA was separated by gel electrophoresis on a 15% TBE-Urea gel

(Invitrogen) and gel fragments extracted corresponding to ~25–35 nt in size. RNA was extracted from gel as in *Ingolia et al. (2011)*.

Harvested cell pellets for mRNAseq were lysed by repeated pipetting in Trizol (Invitrogen) and total RNA isolated per manufacturer protocol. Poly(A) mRNA was purified from the total RNA sample using (dT)$_{25}$ DynaBeads (Invitrogen) per manufacturer protocol. mRNA was fragmented in high pH buffer (50 mM NaCO$_3$, pH 9.2) for 20 min at 95°C, then precipitated and separated by gel electrophoresis as above. mRNA fragments of 50–90 nt were extracted. Footprint and mRNA sample purity and fragment size were verified by Bioanalyzer 2100 (Agilent, Santa Clara, CA, USA).

RNA samples were dephosphorylated, ligated to linker, and separated by gel electrophoresis as described previously (*Ingolia et al., 2011*). Ribosome footprint samples were enriched by subtractive hybridization of contaminating rRNA sequences. We used biotin-modified oligonucletoides and capture on streptavidin-coated beads; oligonucleotide sequences complimentary to rRNA were identical to *Stern-Ginossar et al. (2012)*. Reverse transcription and cDNA library preparation were completed as in *Ingolia et al. (2011)*.

## Analysis of deep sequencing data

Sequencing was performed on an Illumina HiSeq 2000 using single end, 50-bp reads. Before alignment, linker and poly(A) sequences were computationally removed from the 3′ ends of raw sequencing reads. Bowtie v0.12.8 (*Langmead et al., 2009*) (allowing up to two mismatches) was used to perform the alignments with up to two mismatches allowed. For footprint data only reads of length 26–36 nt (footprint length with cycloheximide (*Ingolia et al., 2011*)) were used for alignment. Reads were first aligned vs human rRNA and tRNA sequences; aligned reads were discarded. All remaining reads were next aligned to known canonical transcripts in hg19 (downloaded from genome.ucsc.edu 23 May 2012). Remaining unaligned reads were aligned to the hg19 genome. We used in-house C++ scripts to assign and count unique reads mapping to canonical hg19 transcripts. Only uniquely mapping reads were used for further analysis. The midpoint of footprint reads was used to assign a unique nucleotide location for that read. mRNA-seq reads were assigned a unique nucleotide position at the 5′ end of the read. mRNA and footprint read density were calculated in units of reads per kilobase million (RPKM) to normalize for gene length and total reads per sequencing run. Raw sequencing data is available in the GEO repository with accession number GSE48785.

Selected sequencing reads were visualized through Integrative Genomics Viewer 2.0 (*Robinson et al., 2011*). Metagene analysis was performed as in *Ingolia et al. (2009)*. 5′ UTR read assignments were based on canonical transcript splice isoform as in hg19. For UTR analysis normalized read counts (in RPKM) were used to compare relative 5′ UTR vs CDS translation across time points. For *Figure 1C*, unsupervised hierarchical clustering was performed using centroid linkage across mRNA, footprint, and translation efficiency data with Spearman-Rank correlation in Cluster 3.0 and visualized in TreeView.

All aligned transcripts were initially assigned to a unique UCSC gene ID. Conversion tables available from UCSC through the kgXref function were used to convert UCSC gene ID to HUGO gene nomenclature and to UniProt accession number for further analysis. Where indicated, gene lists were analyzed by Ingenuity Pathway Analysis (Ingenuity Systems, Redwood City, CA, USA) using default settings for included genes and interaction networks.

## iTRAQ proteomics—sample preparation

iTRAQ sample preparation was performed in a similar manner to *Mertins et al. (2012)* with some modifications. 15 × 10$^6$ cells harvested at each time point (as above) were thawed and lysed by sonication in 250 μl lysis buffer (7M Guanidine HCl, 75 mM NaCl, 100 mM bicine pH 8.0) supplemented with protease inhibitors (500 μM 4-(2-Aminoethyl) benzenesulfonyl fluoride HCl, 1 mM E-64, 1 mM phenyl-methylsulfonyl fluoride, and 1 mM EDTA). Lysates were cleared by centrifugation at 16,500×*g* for 10 min and protein concentration was measured by BCA assay. Lysate containing 250 μg of protein was diluted to 100 μl with lysis buffer. Disulfide bonds were reduced with 5 mM dithiothreitol and cysteines alkylated with 10 mM iodoacetimide. Lysate was diluted 1:8 with trypsin dilution buffer (100 mM bicine pH 8.0, 1 mM CaCl$_2$, 75 mM NaCl). Sequencing grade modified trypsin (Promega) was added at an enzyme-to-substrate ratio of 1:25. Trypsin digestion took place for 16 hr with agitation at room temperature. Samples were acidified with trifluoroacetic acid to a final concentration of 0.5% to halt digestion. Tryptic peptides were desalted on Waters SepPak C18 (Milford, MA, USA) columns and evaporated to dryness on a vacuum concentrator.

Peptide samples were resuspended in 40 µl 0.5 M TEAB pH 8.5. Peptide concentration was determined by BCA assay. Labeling with 8-plex iTRAQ reagent was performed per manufacturer instructions (AB Sciex, Framingham, MA, USA): briefly, 100 µg of peptides at each time point was diluted to 30 µl with 0.5 M TEAB; 6-plex iTRAQ reagent was resuspended in 70 µl EtOH. Peptides were mixed with reagent and incubated for 1 hr at room temperature. The reaction was halted using 50 mM Tris/HCl pH 7.5. The labeled reaction products for each sample were combined and evaporated to dryness. Combined labeled peptides were desalted using Waters SepPak C18 cartridge. 0 hr samples were labeled with 113 mass tag; 1.5 hr, 115; 3 hr, 116; 6 hr, 117; 9 hr, 118; 12 hr, 119.

## iTRAQ proteomics–LC-MS/MS

Peptides were resuspended in 0.1% formic acid and then separated into 30 fractions via reverse phase high-pH fractionation using an XBridge C18 column (1.0 × 100 mm, 3.5 µm; Waters) on a Waters 2796 BioSeparations Module HPLC. We used a 70 min gradient with a linear increase from 2% to 38% acetonitrile in water and constant 10% ammonium bicarbonate, pH 10.5; flow rate was 50 µl/min. Fractions were evaporated to dryness and stored at −80°C, then resuspended in 0.1% formic acid prior to mass spectrometry analysis.

The combined iTRAQ sample was then analyzed using two separate mass spectrometers. The first was a QSTAR Elite QqTOF mass spectrometer (AB Sciex) coupled in-line to an Eksigent NLC-1DV-500 HPLC and in-house packed 75-µm × 15-cm $C_{18}$ column. For each fraction in-line LC was performed in Buffer A (0.1% formic acid in water) and Buffer B (0.1% formic acid in acetonitrile) at a flow rate of 250 nl/min. The full method extended over 125 min with a linear gradient from 3% to 40% Buffer B over 90 min, increased to 75% B over 5 min and held for 15 min, then re-equilibration at 3% B for 15 min.

Data-dependent acquisition was performed using Analyst v.2.0 software (AB Sciex) across an $m/z$ range of 385–1200. We used an isolation window of 100 mDa for ions selected for MS/MS with a dynamic exclusion window of 60 s after acquisition. The two most intense ions in each MS1 scan were chosen for CID fragmentation and sequencing. The 'iTRAQ' option was chosen within the instrument software to enhance MS2 collision energy and increase intensity of iTRAQ reporter signal.

Each fraction of the combined iTRAQ sample was also analyzed on an LTQ Orbitrap Velos mass spectrometer (Thermo Fisher Scientific) coupled in-line to a nanoAcquity UPLC system (Waters). Injected samples were trapped on a Symmetry C18 Column (0.18 × 20 mm, 5 µm; Waters) for 5 min at 1% of Buffer B before starting the gradient; an analytical BEH130 C18 column (0.075 × 200 mm column, 1.7 µm; Waters) was used with a flow rate of 600 nl/min. A linear gradient to 38% Buffer B was run over 100 min, then increased to 75% B over 10 min, increased to 90% B over 5 min, then decreased to 2% B and re-equilibrated for 15 min (total method 130 min). Data-dependent acquisition was performed using the Xcalibur 2.1 software in positive ion mode at a spray voltage of 2.5 kV. MS1 survey spectra were acquired in the Orbitrap with a resolution of 60,000 and a mass range from 300 to 1400 $m/z$. All iTRAQ data was acquired by fragmenting the eight most intense ions per cycle in higher-energy collisional dissociation (HCD) mode. Collision energy was set to 45, maximum inject time was 250 ms and maximum ion count was $1 × 10^5$ counts. We used an isolation window of 2.3 Th for ions selected for MS/MS. Ions selected for MS/MS were dynamically excluded for 60 s after acquisition.

## iTRAQ proteomics—peptide identification and protein quantification

Peptide identification was performed using Protein Prospector (v. 5.9) (University of California, San Francisco). All spectra were searched vs the full human SwissProt database (downloaded 21 March 2012) with reverse sequence database as implemented in Protein Prospector for decoy matches. Search parameters included: Fixed modifications carbamidomethyl (cysteine), iTRAQ-8plex (N-terminus), iTRAQ-8plex (lysine); Variable modifications methione-loss (N-terminus) and methionine oxidation; two missed tryptic cleavages allowed; for Orbitrap Velos data, parent and fragment mass tolerance of 20 ppm was used; for QSTAR Elite data, parent mass tolerance was 50 ppm and fragment mass tolerance was 150 ppm. Expectation value thresholds for peptide identification were modified to maintain a false discovery rate <1% based on the number of spectra matching to reverse decoy sequences.

iTRAQ signal was quantified using the Search Compare function of Protein Prospector, which extracts the peak intensity of the MS2 signal in the iTRAQ label range. Only peptides with an iTRAQ

signal >300 cps at the 0 hr time point were included for further analysis to reduce noise in quantification. Only peptides uniquely matching to a single entry in the human SwissProt database were included. Proteins were only included with a minimum of two unique peptides matched. Protein quantification was performed by summation of iTRAQ signal across all peptides assigned to that protein at each time point. Total $\log_2$ iTRAQ signal intensity across all peptides at a time point was median-normalized for comparison across time points.

## SRM assay for iTRAQ validation–method development

We attempted to develop orthogonal, targeted, label-free Selected reaction monitoring (SRM) assays for 250 proteins that were identified in iTRAQ experiments. We primarily concentrated on proteins with relative increases in mRNA expression as found in deep sequencing experiments. Based on prior yeast studies (*Lee et al., 2011*), these would be expected to most likely result in protein changes after perturbation.

We used the open-source software Skyline (v1.3) (*MacLean et al., 2010*) to first build an unscheduled SRM method. Building these methods requires a library of MS/MS spectra for inclusion of the most intense transition pairs (*m/z* data for parent ion and sequence fragment ion) for a given peptide. We used a spectral database of tryptic peptides matched to >8,000 proteins from human cancer cell lines, as identified by extensive fractionation and analysis in HCD mode on the LTQ Orbitrap Velos (AU, J Oses-Prieto, ALB, unpublished data). We selected a minimum of two and up to the seven most intense peptides present in the spectral library for each protein target. For each peptide we used the most intense seven transitions (*y*- and *b*-series ions) for initial measurement by unscheduled SRM. Selected peptides were chosen independently of those found by iTRAQ analysis.

All SRM analysis was carried out on an AB Sciex QTRAP 5500 triple quadrupole mass spectrometer interfaced in-line with a nanoAcquity UPLC system (Waters) identical to that on the LTQ Orbitrap Velos (Trapping column: Symmetry C18 Column (0.18 × 20 mm, 5 µm; Waters); Analytical column: BEH130 C18 column (0.075 × 200 mm column, 1.7 µm; Waters)). We injected ~500 ng of tryptic peptides from MM1.S cells on to the mass spectrometer with the following conditions: Trapping for 4 min in 1% buffer B at 5 µl/min, then after injection a linear gradient from 3%–35% B over 80 min, an increase to 90% B over 5 min, then held for 5 min, then a decrease to 3% B for 10 min (total run time 100 min). Unit resolution was used at Q1 and Q3. A three second cycle time was used for all runs. For unscheduled runs a 10 ms acquisition time was used per transition. Multiple injections were used to test for all targeted peptides. Using data analysis in SkyLine software, peptides were selected for further method development based on (1) the signal detection (above baseline) of at least 5 of 7 co-eluting transitions and (2) a retention time within 7 min of that acquired in the initial spectral library (acquired under the same chromatographic conditions).

Peptides chosen for further development were then limited to the four most intense transitions as found in unscheduled runs. A scheduled SRM method was developed with a retention time window of ±5 min. The four most intense transitions were first subject to collision energy (CE) optimization in ±4 V steps to enhance intensity of detected transitions. CE-optimization data from QTRAP 5500 on the same samples was again analyzed in SkyLine. Peptides were confidently assigned based on the identification of four out of four targeted transitions co-eluting at the appropriate retention time. Ultimately, our final SRM method included from one to three confidently-assigned peptides matching to 152 proteins (all transitions listed in *Figure 4—source data 1*) with the CE for each transition used which maximized detected intensity.

## SRM assay for iTRAQ validation—sample analysis

We prepared an independent time course of MM1.S cells exposed to 20 nM bortezomib and harvested at 0 hr, 1.5 hr, 3 hr, 6 hr, 9 hr, and 12 hr. Cell viability and caspase activity assays were similar to that shown in *Figure 1B*. We prepared tryptic peptides from 15 × 10$^6$ cells at each time point (as described above prior to iTRAQ labeling). ~500 ng of tryptic peptides were injected in triplicate and analyzed using the final scheduled SRM assay.

Peptide intensity in each sample was measured as the sum of all transition peak areas for that peptide (as measured by analysis in SkyLine). For peptides not clearly identified at all time points, a retention time window was used identical to that in other injections with background signal integrated and used for analysis. To normalize peptide concentration across samples, we used peptides derived from a set of high abundance proteins not expected to significantly change during the time course

(actin, tubulin, and filamin-A). We derived an index based on the geometric mean intensity of peptides from these 'housekeeping' proteins and scaled SRM intensity of all peptides in each sample based on the median value of this index. Corrected peptide intensity was averaged across injections for each sample. Protein intensity was measured as the sum of peptide intensity for each sample.

## Estimation of absolute mRNA and protein copies per cell

For mRNA copy number estimation we used an approach similar to that used in *Schwannhausser et al. (2011)*, described initially by *Mortazavi et al. (2008)*. Total mRNA measured in duplicate by NanoDrop in the untreated MM1.S sample ($0.88 \pm 0.09$ µg) was divided by $4 \times 10^6$ cells to estimate the mRNA yield per cell ($2.19 \times 10^{-13}$ g). mRNA sequencing reads were approximately evenly distributed across A:C:G:T; the average molecular weight of a RNA monophosphate nucleotide (averaged across AMP, CMP, GMP, and UMP) is 339.5 g/mol. We can thus calculate the total number of mRNA nucleotides per cell ($T$; $6.40 \times 10^{-15}$ mol/cell). The copy numbers of individual mRNAs ($c$) can be calculated by the total number of sequencing reads which mapped uniquely to a given transcript ($r_{transcript}$), the total reads ($r_{total}$; 8,744,024 reads uniquely aligning to hg19 transcripts for the 0 hr MM1.S sample), and the transcript length ($L$) by the equation:

$$\frac{r_{transcript}}{r_{total}} = \frac{c \times L}{T}.$$

For protein copy number estimation, we harvested and prepared tryptic peptides (as described above prior to iTRAQ labeling) from $15 \times 10^6$ untreated MM1.S cells. Peptides were separated into 20 fractions by reverse-phase high pH chromatography (same protocol as above for iTRAQ peptides). Each fraction was analyzed separately in HCD mode on the LTQ Orbitrap Velos Mass Spectrometer and nanoAcquity UPLC as above (gradient: 60 min linear gradient from 3% to 30% buffer B, 5 min gradient to 90% B, held 10 min, re-equilibrate 15 min at 3% B; mass spectrometer settings same as for iTRAQ analysis except collision energy at 35% and ion accumulation at $1 \times 10^6$). We analyzed the MS data using the MaxQuant software package (v.1.3.0.5) (*Cox and Mann, 2008*) with Carbamidomethyl cysteine as a fixed modification; Methionine oxidation and N-terminal acetylation as variable modifications; initial parent and fragment MS/MS tolerance of 20 ppm; a minimum peptide length of seven; one missed tryptic site; and razor peptides used for iBAQ quantification. Peptides were searched against the full Uniprot Human Proteome database (downloaded November 10, 2012) and contaminants enabled in MaxQuant; maximum allowable peptide and protein false discovery rate was 1% as searched against reverse sequence library based on Uniprot database. In MaxQuant peptide matches were assigned to protein groups (a cluster of a base protein plus additional proteins matching to a subset of the same peptides). Protein groups matching the reverse database or contaminants were discarded.

iBAQ quantification in MaxQuant evaluates the intensity of each protein in the sample as the sum of all the peptide intensities in the MS1 scan for peptides which matched to that protein group. iBAQ analysis and database searching resulted in assignment of intensity to 3390 protein groups with a minimum of one assigned peptide. We assigned each protein group to an individual protein based on the primary UniProt ID associated with the analysis output. We used the total iBAQ signal in the sample ($I_{total}$), across all matched peptides, as a measure of the total cellular protein signal as measured on the mass spectrometer. We divided the measurement of total protein in the sample (from BCA assay) by $10 \times 10^6$ cells to obtain an estimate of total protein per cell ($g_{total}$; $5.28 \times 10^{-11}$ g). We then calculated the grams of protein in the cell comprised of an individual protein ($g_{protein}$) based on the iBAQ signal assigned to that protein ($I_{protein}$) as a fraction of total iBAQ signal.

$$\frac{g_{protein}}{g_{total}} = \frac{I_{protein}}{I_{total}}$$

Using the grams of an individual protein per cell, the molecular weight of the assigned protein in g/mol, and Avagadro's number, we assign an estimated number of copies per protein per cell.

These measures of absolute copies of mRNA transcripts and proteins per cell are clearly rough approximations. In an important initial comparison, the median of the distributions for both mRNA and protein copy per cell are of the same order of magnitude as, though somewhat lower than, those

found in *Schwanhausser et al. (2013)*. This may reflect differences in the cell line (mouse NIH3T3 vs human MM1.S) as well as differences in measurement. However, the overall similarity of the distributions found here to those found by *Schwanhausser et al. (2013)* indicates that our data are sufficient as a guide for our mass-action model (below). We further emphasize that our data are not intended as a definitive measure of mRNA or protein copy numbers in MM1.S cells. Even if there are systematic errors in the quantitation (i.e., if true protein copies per cell are all threefold higher), the fitting parameters for our mass-action model (see below) will be uniformly scaled but the overall interpretation of the model will be unchanged. This issue of scaling was illustrated in the corrigendum to the original manuscript by *Schwanhausser et al. (2013)*. The original paper (*Schwanhausser et al., 2011*) used the incorrect standard for protein abundance estimation, leading to threefold in changes in the protein distribution; the corrected scale led to an increase in the fit parameter $k_{sp}$ but no changes to the model or conclusions of the paper.

## Quantitative modeling of protein translation

To model the conversion of mRNA to protein in this system we used a mass-action model of translation, as originally described in *Hargrove and Schmidt (1989)*, and further used by *Schwanhausser et al. (2011)*, as an ordinary differential equation:

$$\frac{dP}{dt} = k_{sp}R(t) - k_{dp}P(t)$$

Here the number of protein copies ($P(t)$) changes over time as a function of the number of mRNA transcripts ($R(t)$), the translation rate constant $k_{sp}$, and the protein degradation rate constant $k_{dp}$.

In our system, we first modeled $R(t)$ using the normalized sequencing reads (RPKM) at later time points and taking the ratio to the RPKM at 0 hr. We then multiplied this RPKM ratio at each time point by the absolute number of mRNA transcripts at 0 hr as measured above. We focused on analyzing only a subset of 13 transcripts with increased relative mRNA abundance but with variable detected changes at the protein level (i.e., some with increased protein, some with no protein change). We plot this data in *Figure 5D*, *Figure 5—figure supplement 4A,C*. We found that for this subset of transcripts, this mRNA transcript data could be well-approximated by fitting either a quadratic (FTL, SQSTM1, CTSD) or 4-parameter sigmoid (all others). This fit function (fitting performed in GraphPad Prism software) describes $R(t)$.

However, as discussed in the main text, this function $R(t)$ is based on *relative* transcript expression at different time points. This would be an effective measure of mRNA transcripts per cell in the setting of constant total mRNA concentration. In this system, though, we find that mRNA is significantly degraded at later time points (*Figure 1B*). Therefore, we fit an additional function $G(t)$ to this mRNA data as a four-parameter sigmoid fit to describe this global mRNA degradation. By incorporating $G(t)$ into the mass-action model above, assuming similar degradation rates of each transcript during apoptosis (as described by *Del Prete et al. (2002)*), the combined term $R(t)G(t)$ now represents the *absolute* number of mRNA transcripts per cell in our system:

$$\frac{dP}{dt} = k_{sp}R(t)G(t) - k_{dp}P(t).$$

To estimate absolute protein copies per cell at later time points, we used the label-free SRM intensity data (as this does not suffer from compression ratio artifacts as seen in iTRAQ data). We took the ratio of SRM intensity at later time points to that at 0 hr, then multiplied by the absolute copies protein copies per untreated MM1.S cell as measured by iBAQ. We scaled the 12 hr SRM data to reflect the decreased protein concentration in the sample.

We used Mathematica (v. 9) software to numerically solve the differential equation above for each of the 13 genes chosen for analysis. We fit this differential equation to the plotted protein data (as shown in *Figure 5E*, *Figure 5—figure supplement 4B,D*). This equation has three free fitting parameters to describe the six protein data points: protein copies at 0 hr, $k_{sp}$, and $k_{dp}$. Protein copies at 0 hr was varied a maximum of 10% from the measured protein copies per cell by iBAQ. $k_{dp}$ was assumed to be relatively constant for all proteins in the background of proteasomal blockade by bortezomib. We only varied the range between 0.010–0.015 hr$^{-1}$, similar to the corresponding

average protein half-life found in *Schwanhausser et al. (2011)*. Therefore, we primarily varied the translation rate constant $k_{sp}$ in order to fit this model to the protein data. We performed the fits by manually minimizing the least-squares difference between model fit and protein data in Mathematica. We found that the model could indeed well-describe the protein data; in addition, the fit $k_{sp}$ parameter ranged from 10 to 270 proteins/transcript/hr, consistent with the range in *Schwanhausser et al. (2011)*.

### SRM assay for degradomics

As described in *Shimbo et al. (2012)*, cell pellets were lysed by sonication in 4.0% SDS, 400 mM Bicine (pH8.0), and supplemented with protease inhibitors 0.1 mM z-VAD-fmk, 0.1 mM E-64, 1 mM AEBSF, 1 mM PMSF, and 5 mM EDTA. Spike-in internal standards were added to each sample after lysis: 50 µg each of bovine catalase and yeast alcohol dehydrogenase (Sigma). Proteins were reduced by TCEP, cysteines alklyated by iodoacetamide, and free N-termini biotinylated via the reaction of TEVest4 ester and subtiligase enzyme as previously described (*Shimbo et al., 2012*). Biotinylated peptides were precipitated, resuspended in 5.3 M Guanidine HCl and captured on NeutraAvidin agarose beads (Thermo). The beads were extensively washed in 5 M Guanidine HCl followed by on-bead trypsin digestion overnight at room temperature with agitation. As previously described, captured peptides were released from beads by incubation with TEV protease (*Shimbo et al., 2012*). Peptides were desalted using $C_{18}$ ZipTips (Millipore), evaporated to dryness, and stored at −80°C for analysis.

SRM assays were developed in a similar manner as described above, again using SkyLine software. In this case, however, targeted parent/fragment ion transitions were extracted from MS/MS spectra in the DegraBase. This publicly available database lists N-terminally labeled peptides found in human cell culture under both apoptotic and non-apoptotic conditions (wellslab.ucsf.edu/degrabase). *b*-ion fragment $m/z$ was modified as necessary to reflect the Abu- tag at the N-terminus of the peptide (*Shimbo et al., 2012*). We initially chose 400 peptides for development of targeted SRM assays in SkyLine. We used the same chromatography and instrument settings as described for the earlier SRM assays. We initially examined seven transitions per peptide; we moved forward with peptides with a minimum of five co-eluting transitions and a retention time within 7 min of that predicted based on the peptide retention time prediction algorithm (SSRCalc 3.0) implemented in SkyLine. Ultimately, we included 252 proteolytic peptides in the final SRM assay.

We corrected samples for labeling efficiency and overall peptide concentration using five proteolytic peptides derived from the spike-in standards. Corrected peptide intensities were averaged across duplicate injections at each time point in the 20 nM bortezomib time course. These mean intensity values were used for analysis.

## Additional information

### Funding

| Funder | Grant reference number | Author |
| --- | --- | --- |
| National Institutes of Health | R01GM081051 | James A Wells |
| Howard Hughes Medical Institute | | Jonathan S Weissman |
| Damon Runyon Cancer Research Foundation Post-Doctoral Fellowship (DRG 111-12) | | Arun P Wiita |

The funders had no role in study design, data collection and interpretation, or the decision to submit the work for publication.

### Author contributions

APW, Conception and design, Acquisition of data, Analysis and interpretation of data, Drafting or revising the article; EZ, Acquisition of data, Analysis and interpretation of data; PJW, OJ, Analysis and interpretation of data, Drafting or revising the article; AU, Conception and design, Contributed unpublished essential data or reagents; ALB, Drafting or revising the article, Contributed unpublished essential data or reagents; JSW, JAW, Conception and design, Drafting or revising the article

# Additional files

## Major dataset

The following dataset was generated:

| Author(s) | Year | Dataset title | Dataset ID and/or URL | Database, license, and accessibility information |
|---|---|---|---|---|
| Wiita AP, Ziv E, Wiita PJ, Urisman A, Julien O, Burlingame AL, et al. | 2013 | Global response to chemotherapy-induced apoptosis | GSE48785; http://www.ncbi.nlm.nih.gov/geo/query/acc.cgi?acc=GSE48785 | Publicly available at GEO (http://www.ncbi.nlm.nih.gov/geo/). |

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
