## [Decision Letter]

[Editors’ note: although it is not typical of the review process at *eLife*, in this case the editors decided to include the reviews in their entirety for the authors’ consideration as they prepared their revised submission.]

Thank you for sending your work entitled “Global cellular response to chemotherapy-induced apoptosis” for consideration at *eLife*. Your article has been favorably evaluated by a Senior editor, a Reviewing editor, and 2 reviewers.

The Reviewing editor and the two reviewers discussed their comments before we reached this decision, and the Reviewing editor has assembled the following comments to help you prepare a revised submission.

Both reviewers were enthusiastic about your manuscript and its findings. However, the reviewers have also recommended some revisions to strengthen the paper. Specifically, questions were raised about whether a more thorough bioinformatics analysis, especially one that better integrates the ribosome profiling and proteomics data sets, might identify important examples where protein levels were controlled primarily by proteolytic degradation (or strongly affected by unconventional 5' UTRs). To the extent that you and your co-authors can address these interesting questions, we believe that the revised manuscript would be further improved.

In this case we think it would be beneficial to also include the full reviews of your article as you prepare your revision.

*Reviewer 1*:

Overall, this is an interesting study that uses multiple different types of profiling technologies to establish how multiple myeloma cell lines response to bortezomib, the common anti-multiple myeloma treatment.

The authors characterize a variety of different transcriptional and post transcriptional responses, including changes in transcript levels, changes in ribosome profiling read density, protein levels, and protease cleavage.

Overall, the manuscript is interesting because of the multiple different approaches.

If there is a flaw, it is that many of the bioinformatics analyses seem to be performed without a strong rationale (or at least a stated rationale). The data seems to be available, so the data is analyzed. As such, in many cases, no clear hypotheses are generated.

Here are a few examples:

1) The authors perform both ribosome profiling and proteomics (iTRAQ). But the authors say that the purpose of this experiment is to measure “the production of proteins during bortezomib-induced apoptosis.” I don't think the authors are properly considering their iTRAQ data, especially with respect the to ribosome profiling data. iTRAQ does not speak to protein production, it speaks to overall protein levels, which is a combination of protein synthesis and degradation. Ribosome profiling gives protein production. The proteomic data would be more valuable if the authors would contrast this with the ribosome profiling data, and then see if the differences are due to proteolytic degradation. For instance, some proteins may be found at a lower level than predicted by the ribosome profiling. Are these proteins that are targeted for degradation?

2) The authors devote a section to changes in 5'UTR reads. However, they don’t define what these 5'UTR reads correspond to. I think they correspond to uORFs and unconventional in-frame start sites upstream of the canonical ATG. If these are the only two types of 5'UTRs, it seems that these should be analyzed separately since they might have different forms of regulation.

3) Along the lines of the 5'UTR reads: It seems to me that the model is that increased 5'UTR reads might lead to lower CDS reads, and thereby contribute to the overall protein translation level seen by ribosome profiling and overall expression level seen by iTRAQ. But, this doesn’t seem to be bioinformatically tested. It seems that the 5'UTR reads are analyzed without seeing how these are influencing overall protein levels.

4) I am not sure if I can tell if the N-terminalomic data gives insights into whether the cleavage events are rare or at a high stoichiometry. Perhaps the authors can comment on this. It seems to me that the authors could extrapolate the important proteolytic cleavage events by looking at the ribosome profiling and iTRAQ datasets (as discussed in point 1 above) to possibly address this issue.

*Reviewer 2*:

The study of Wiita et al. reports the systems-level analyses of transcription, translation and proteolysis in the multiple myeloma cancer cell line treated with a proteasomal inhibitor bortezomib. This study represent an unprecedented integrative study and even though the conclusive findings are not world shocking, the use of state-of-the-art technologies, the experimental design and data integration really makes this a reference work for future integrative studies and puts this study in the class of pioneering integrative -omics studies.

Further, the authors highlight some of the pitfalls, such as the increased ionization suppression of iTRAQ reporter ions as compared to SRM approaches in mass spectrometry, which are typically overlooked when solely making use of one -omics approach.

Further since (regulated) protein proteolysis/degradation cannot be studied by means of ribosome profiling, the combined use of ribosome profiling and shotgun/N-terminomics approaches additionally enables the study of protein degradation; in this case mainly reflected by caspase-mediated proteolysis.

Since this study represents a comprehensive and integrative -omics story, and viewing the fact that RNA-Seq and proteomics data from the same model system was obtained, the creation of a customized database could have been considered for the comprehensive identification/discovery of database non-annotated peptides (i.e., splice-junction peptides, products of uORF translation, translation at near-cognate start codons, ...). The authors could try to implement this or at least discuss some of the advantages using such an approach and refer to some previous studies making use of customized databases for proteomics (62; 42; 58).

Overall, this study is well designed and the manuscript well written and this reviewer strongly supports publication of this important work.

---

## [Author Response]

Response to Reviewer 1:

*1) The authors perform both ribosome profiling and proteomics (iTRAQ). But the authors say that the purpose of this experiment is to measure “the production of proteins during bortezomib-induced apoptosis.” I don't think the authors are properly considering their iTRAQ data, especially with respect the to ribosome profiling data. iTRAQ does not speak to protein production, it speaks to overall protein levels, which is a combination of protein synthesis and degradation. Ribosome profiling gives protein production. The proteomic data would be more valuable if the authors would contrast this with the ribosome profiling data, and then see if the differences are due to proteolytic degradation. For instance, some proteins may be found at a lower level than predicted by the ribosome profiling. Are these proteins that are targeted for degradation*?

We thank the reviewer for the astute comment. It is absolutely true that the iTRAQ data reflects overall protein levels and, in general, not synthesis alone. In our experimental system, however, treatment with a high dose of bortezomib effectively blocks proteasomal degradation in this system. Bortezomib’s IC_50_ for proteasome inhibition in MM1.S cells is known to be ∼5 nM (Chauhan D et al., *Cancer Cell* (2005) 8, 407) and the 20 nM dose has previously been shown to lead to essentially complete proteasomal blockade (Berkers CR et al., *Nat Methods* (2005) 2, 357). Conversely, other endoproteases (such as caspases) appear to cleave protein substrates in a limited number of sites generating large protein fragments (Crawford et al. *Mol Cell Proteomics* (2013) 12, 813; Dix et al *Cell* (2008) 134, 679; Mahrus et al. *Cell* (2008), 134, 866). In the absence of proteasomal activity, these fragments would be expected to remain stable in the cell as large polypeptides (whether retaining any function or not). By iTRAQ mass spectrometry, based on trypsinization of the total protein sample, these proteolytically cleaved fragments would be indistinguishable from intact proteins. As a result iTRAQ provides no information on caspase (or non-caspase) proteolysis which does not degrade proteins below the domain level. Therefore, in this system we believe that degradation as measured by iTRAQ would both be close to zero for all proteins, and relative iTRAQ data across time points largely reflects protein production.

To clarify this point we have changed the line referred to by the reviewer to “changes in protein levels during bortezomib-induced apoptosis”. We have also modified another statement to state “at 20 nM bortezomib in MM1.S cells, proteasomal activity is almost completely inhibited (6)” and added a statement later in the same paragraph: “These domains would be indistinguishable from intact proteins by iTRAQ. Therefore, we expect that protein degradation detectable by iTRAQ is extremely limited and changes by iTRAQ largely reflect protein production alone.”

The reviewer further makes an excellent point that ribosome profiling data could be used as a predictor for protein production. One could subsequently examine discrepancies between predicted protein production and that detected by proteomics. For the quantitative model presented in Figure 5, the ribosome profiling data is implicitly included as none of the genes analyzed had large changes in translational efficiency. Hence, any protein changes predicted by ribosome profiling would be equivalent to changes predicated by mRNA-seq data. As shown in Figure 5—figure supplement 4, this model well-describes the analyzed genes and does not lead to any notable discrepancies.

However, a different and interesting question is inferring protein abundance changes directly from ribosome profiling data (i.e., X number of increased ribosome footprints = Y number of new proteins). We previously attempted to incorporate this analysis as another element of our quantitative model, but theoretical studies suggest that the predicted protein output of a transcript is not a straightforward function of measured ribosome footprint read density (Tuller T. et al. *Cell* (2010) 141, 344; Cannarozzi G. et al. *Cell* (2010) 141, 355). In our current study under conditions of rapid apoptosis, we were unable to test or develop any new predictions for converting ribosome occupancy to protein synthesis as so few proteins show detectable changes in iTRAQ data.

*2) The authors devote a section to changes in 5'UTR reads. However, they don’t define what these 5'UTR reads correspond to. I think they correspond to uORFs and unconventional in-frame start sites upstream of the canonical ATG. If these are the only two types of 5'UTRs, it seems that these should be analyzed separately since they might have different forms of regulation*.

The reviewer is correct that footprint reads present in 5’ UTRs could apply to either translated upstream open reading frames (uORFs) generating short polypeptides separate from the coding sequence or N-terminal extensions leading to different isoforms of the canonical protein. Indeed we did not clearly define this in the main text. As we mention in the text, stalling translation initiation complexes with the drugs harringtonine or lactimidomycin is necessary to definitively identify start sites. Using the translation elongation inhibitor cycloheximide, as we do here, we cannot readily define these different types of 5’ UTR translation across all transcripts and we therefore must analyze them together. However, to the reviewer’s point, we have added a line to the text stating that “Ribosome occupancy in this region may indicate translation of short regulatory polypeptides in upstream open reading frames (uORFs) or alternate production of N-terminal isoforms of canonically translated proteins...” In the few cases we examine in detail (Figure 3, Figure 3—figure supplement 1, Figure 3—figure supplement 2 and Figure 3—figure supplement 3), we characterize these as uORFs because all of the analyzed translation products, whether in-frame or out of frame, have a stop codon prior to the canonical coding sequence; N-terminal extensions would both be in-frame with the canonical coding sequence and have no intervening stop codon.

*3) Along the lines of the 5'UTR reads: It seems to me that the model is that increased 5'UTR reads might lead to lower CDS reads, and thereby contribute to the overall protein translation level seen by ribosome profiling and overall expression level seen by iTRAQ. But, this doesn’t seem to be bioinformatically tested. It seems that the 5'UTR reads are analyzed without seeing how these are influencing overall protein levels*.

We thank the reviewer for this suggestion to integrate our 5’ UTR analysis with proteomic data. We point out in Figure 3 that our current analysis compares the ratio of read density in the 5’ UTR to the protein coding sequence (CDS) across the time course. We did not see a strong relationship between changes in UTR translation affecting CDS translation. In support of our result, Geraschenko et al. (*PNAS* (2012) 109, 17394) noted large increases in 5’ UTR translation in yeast after oxidative stress but noted a similarly complex relationship to translation of the protein coding sequence. So from this analysis it does not appear that increased 5’ UTR translation always leads to decreased CDS translation.

To further explore this idea, we have performed two additional pieces of analysis. We first compared whether relative 5’ UTR translation in untreated cells shows a relationship to steady-state protein abundance by iBAQ. In this analysis, incorporating 2266 genes with 5’ UTR reads and iBAQ data, we visually observe a negative relationship: genes with high levels of relative 5’ UTR to CDS translation appear to show lower levels of protein abundance (new Figure 5—figure supplement 2; new source data in [Supplementary-material SD4-data]). However, statistical analysis demonstrates a non-significant negative correlation (Pearson *R* = -.02, *p* = 0.35). Therefore, we cannot definitively conclude that increased 5’ UTR translation at baseline leads to lower protein abundance. We have now included a statement: “Interestingly, we also found that increased 5’ UTR translation at baseline may correlate with decreased steady-state protein abundance (Figure 5—figure supplement 2).”

We further examined the changes in relative 5’ UTR read density across the few proteins that are found to be increased by iTRAQ analysis across a time course. We also compared these results to a set of genes with increased read density at the mRNA and ribosome footprint data but no detected change by iTRAQ (new Figure 4—figure supplement 3). The 5’ UTR to CDS read density remained relatively constant for all genes across the time course. We did not, for instance, note consistent decreases in 5’ UTR translation in the genes with increased protein abundance. Instead, this ratio showed little divergence from the value at 0h, consistent with our analysis of all genes in Figure 3. However, as mentioned above, this analysis is necessarily limited by the very few proteins for which we saw abundance changes by iTRAQ. We have added a statement: “Of note, we did not observe any distinct changes in relative 5’ UTR translation for the few proteins increased by iTRAQ (Figure 4—figure supplement 3).”

*4) I am not sure if I can tell if the N-terminalomic data gives insights into whether the cleavage events are rare or at a high stoichiometry. Perhaps the authors can comment on this. It seems to me that the authors could extrapolate the important proteolytic cleavage events by looking at the ribosome profiling and iTRAQ datasets (as discussed in point 1 above) to possibly address this issue*.

The reviewer is correct that the N-terminomic data over a time course, while providing useful information on relative substrate cleavage, does not determine the absolute amount of substrate cleavage relative to the total protein present. As mentioned above in the response to point 1, the iTRAQ data unfortunately also does not offer much insight into protein degradation in this system. This is further evidenced by the fact that so few proteins show signatures of decreased abundance by iTRAQ (Figure 4) whereas many appear to be proteolytically cleaved by SRM (Figure 6). We gain some insight into stoichiometry based on Western blotting results (as in Figure 5 for PERK and 4E-BP1 and Figure 1—figure supplement 2 for XIAP and ATF-4), suggesting that peptides which show strong proteolytic cleavage by SRM indicate ∼50 to >90% of total protein cleaved. This extent of cleavage has also been seen by Western blotting in our prior studies incorporating SRM (Agard NJ et al. *PNAS* (2012) 109, 1913; Shimbo K et al *PNAS* (2012) 109, 12432).

Response to Reviewer 2:

*Since this study represents a comprehensive and integrative -omics story, and viewing the fact that RNA-Seq and proteomics data from the same model system was obtained, the creation of a customized database could have been considered for the comprehensive identification/discovery of database non-annotated peptides (i.e., splice-junction peptides, products of uORF translation, translation at near-cognate start codons, ...). The authors could try to implement this or at least discuss some of the advantages using such an approach and refer to some previous studies making use of customized databases for proteomics (62; 42; 58)*.

The reviewer makes an excellent point that customized databases incorporating ribosome profiling data for proteomic searches are likely to have a significant impact on future studies. We are aware of the references cited but unfortunately our current data does not lend itself to similar analysis as we cannot definitively identify uORF and alternate start sites in our data, as discussed above. In addition, regarding prediction of splice-junction peptides, these data are greatly informed by the use of long reads and paired-end mRNA-seq to enhance alignment across splice junctions. Our data are currently composed of short reads (50 bp) and single-end sequencing which cannot lead to as comprehensive or accurate splice junction assignment. Therefore, we do not believe that our current data set is appropriate for the creation of such a reference database, though it is clear that such databases are certainly important. We have added a comment to the Discussion relevant to the reviewer’s point: “Incorporation of translation initiation inhibitors in ribosome profiling and paired-end reads in mRNA-seq can also provide novel insight into translation regulation through generation of custom proteomic databases (62; 42; 58).”